# A Simple Linear Patch Revives Layer-Pruned Large Language Models

**Xinrui Chen**[1]*, **Haoli Bai**[2]*, **Tao Yuan**[2], **Ruikang Liu**[1], **Kang Zhao**[2], **Xianzhi Yu**[2],
**Lu Hou**[2], **Tian Guan**[1], **Yonghong He**[1], **Chun Yuan**[1]*

[1]Shenzhen International Graduate School, Tsinghua University
[2]Huawei Technologies
baihaoli@huawei.com, yuanc@sz.tsinghua.edu.cn

## Abstract

Layer pruning has emerged as a widely used technique for compressing large language models (LLMs). However, existing layer pruning approaches often incur substantial performance degradation. We identify the majority of this degradation to a single yet previously overlooked issue: *the mismatch of activation magnitudes at the pruning interface*. The pre-interface activations exhibit significantly different scales from the post-interface ones, causing the distributional shift as it propagates through the remaining layers. To address this issue, we introduce LINEARPATCH, a lightweight and plug-and-play technique that fuses two operations into one matrix multiply at the pruning interface: (i) a Hadamard transformation that suppresses massive outliers at particular tokens and (ii) a channel-wise scaling that aligns activation statistics. On LLaMA-3-8B, LINEARPATCH preserves up to **94.15%** of the original model's performance when pruning 5 out of 32 layers, outperforming the previous state of the art by **4%**. The patch can be further refined with 5K unlabeled samples via memory-efficient offline distillation, pushing the retention to 95.16% within only 30 minutes on a single GPU. Code is available at https://github.com/chenxinrui-tsinghua/LinearPatch.

## 1 Introduction

Recent large language models (LLMs) have achieved remarkable progress toward artificial general intelligence [2, 25, 55, 16, 46, 31, 19, 49, 50]. However, their massive scale also incurs substantial computational and memory costs during deployment. To address this issue, a variety of model compression techniques have been proposed, including quantization [5, 52, 4, 45, 10, 33, 29] and pruning [3, 48, 51, 40, 24, 13].

Among these, *layer pruning* has emerged as an attractive approach since it can be readily applied without relying on hardware-specific optimizations or low-level kernel modifications [42, 26, 8, 37, 18]. In contrast, unstructured pruning [20, 17, 44] is difficult to accelerate efficiently due to irregular memory access patterns, while structured sparsity [3, 48] and N:M sparsity [17, 57] often require modifications to model architectures or specialized kernels. Layer pruning, on the other hand, simply removes redundant layers without additional dependencies. However, despite the simplicity, a critical challenge is the sharp drop of performance.

In this work, we uncover a new phenomenon that explains this degradation: *the mismatch of activation magnitudes across layers and tokens at the pruning interface*. Specifically, when layers are pruned, the activations from the remaining layers often exhibit different scales, and the activations from the layer preceding the pruning interface may not align with those from the subsequent layer. This

---

*Corresponding authors.

39th Conference on Neural Information Processing Systems (NeurIPS 2025).

mismatch is further exacerbated by the presence of *massive outliers* in the activations of special tokens (e.g., [BOS] or delimiter tokens), which are commonly observed in LLMs [32, 43]. As a result, pruned LLMs suffer from severe activation mismatch, ultimately leading to performance degradation.

To address this issue, we propose LINEARPATCH, a plug-and-play technique designed to compensate for activation mismatches. The method can be seamlessly integrated with various pruning metrics. Concretely, LINEARPATCH first applies a Hadamard transformation to suppress massive outliers in the activations of special tokens [32, 43]. It then introduces a channel-wise scaling parameter to bridge the gap in activation magnitudes. *Leveraging spectral theory, both the Hadamard transformation and the diagonalized channel-wise scaling can be combined into a real symmetric matrix, which we insert at the pruning interface as* LINEARPATCH. This approach incurs negligible inference overhead while effectively aligning activation magnitudes. Beyond alignment, we further enhance pruned LLMs through memory-efficient knowledge distillation. In particular, we fine-tune only the LINEARPATCH matrix while freezing all other model parameters. This efficient training process requires only 5K samples and can be completed within 30 minutes on a single GPU for a 7B model.

Our empirical results demonstrate the effectiveness of LINEARPATCH across diverse LLMs and tasks. For example, on the question answering benchmark, LINEARPATCH preserves up to **94.15%** of the performance when pruning 5 layers from LLaMA-3-8B, significantly outperforming state-of-the-art methods such as LLM-Streamline (90.84%). Moreover, with our efficient knowledge distillation, the retained performance can be further boosted to **95.16%**. These results highlight LINEARPATCH as a simple yet powerful solution for reviving layer-pruned LLMs with minimal overhead.

## 2 Related Work

**Weight Pruning.**   Weight pruning for LLMs can be categorized into *structured* and *unstructured* pruning, depending on the regularity of the pruning pattern. Unstructured pruning removes individual weights based on their importance, without considering structural organization. A pioneering work [20] prunes weights with the smallest magnitudes, followed by retraining to restore accuracy. Wanda [44] extends this idea by pruning weights according to the product of their magnitudes and the corresponding input activations, outperforming standard magnitude-based pruning. While unstructured pruning is relatively straightforward and can achieve higher compression ratios, it cannot be efficiently accelerated due to irregular memory access.

In contrast, structured pruning eliminates entire groups of weights, thereby preserving computational efficiency. In addition, *N:M* sparsity only keeps *N* elements out of every *M* contiguous weights, which yields more flexibility compared to structured pruning [17, 57]. However, such approach typically requires architectural modifications or customized low-level kernels for acceleration. Beyond *N:M* sparsity, other structured pruning techniques typically remove entire groups of parameters, such as attention heads, MLP neurons, or hidden dimensions [3, 48, 51, 40, 24]. Although more hardware-friendly than unstructured pruning, width pruning still introduces structural irregularities and inevitable requires re-training.

**Layer Pruning.**   More recently, *layer pruning* has emerged as a promising direction for compressing LLMs. Unlike width pruning, which often produces irregular architectures, layer pruning removes entire Transformer layers—including both attention and MLP modules, making it easier to deploy and accelerate. A series of recent works have explored this direction. ShortGPT [37] evaluates layer importance via cosine similarity between layer inputs and outputs, pruning the least critical layers. SLEB [42] iteratively prunes redundant layers by measuring perplexity degradation on a calibration set. Shortened LLaMa [26] explores Taylor-based metrics and perplexity as pruning metrics, and employs LoRA (Low-rank adaptation) fine-tuning to recover accuracy. UIDL [18] introduces an angular distance metric to identify and remove consecutive layers, followed by QLoRA fine-tuning to mitigate accuracy loss. LLM-Streamline [8] identifies the least important consecutive layers using cosine similarity and replaces them with a lightweight network, reporting that fine-tuning this lightweight network with MSE loss outperforms LoRA fine-tuning.

Despite their effectiveness, these approaches consistently overlook a critical issue: the *magnitude mismatch* that arises at the pruning interface. We demonstrate that this mismatch is highly detrimental to performance. In contrast, our proposed LINEARPATCH explicitly addresses this challenge by

introducing a lightweight yet effective magnitude-alignment patch, achieving superior performance under both training-free and post-training settings compared with existing layer pruning methods.

## 3 Method

### 3.1 Preliminaries on LLM Layer Pruning

LLMs primarily adopt the Transformer architecture, consisting of a stack of Transformer decoder layers, each with a residual structure. We denote the $\ell$-th Transformer layer as $f(\mathbf{X}^{(\ell)}; \theta^{(\ell)})$, where $\mathbf{X}^{(\ell)}$ and $\theta^{(\ell)}$ represent its input activations and parameters, respectively. Under the prevalent pre-norm architecture, the input to the $(\ell + 1)$-th layer can thus be obtained by:

$$\mathbf{X}^{(\ell+1)} = \mathbf{X}^{(\ell)} + f(\mathbf{X}^{(\ell)}, \theta^{(\ell)}). \tag{1}$$

**Pruning Metrics.** To identify redundant layers, several metrics are commonly used, including cosine similarity [37, 18, 8], gradient-based scores [26, 35], and perplexity-based scores [26, 42]. For example, LLM-Streamline [8] removes $n$ contiguous layers with the highest cosine similarities among their input activations. The optimal pruning index $\ell^*$ is determined by:

$$\ell^* = \arg\max_{\ell} \mathbb{E}_{(\mathbf{X}_i^{(\ell)}, \mathbf{X}_i^{(\ell+n)}) \in \mathcal{D}} \frac{\mathbf{X}_i^{(\ell)} \cdot \mathbf{X}_i^{(\ell+n)}}{\|\mathbf{X}_i^{(\ell)}\| \|\mathbf{X}_i^{(\ell+n)}\|}, \tag{2}$$

where $\mathcal{D}$ denotes the calibration set used to compute layer activations, and $\mathbf{X}^{(\ell)} \in \mathbb{R}^{B \times L \times C}$ is the input of the $\ell$-th layer, with batch size $B$, sequence length $L$, and hidden dimension $C$.

**Layer Pruning.** Suppose we remove $n$ consecutive layers starting from $\ell^*$. The $(\ell^* + n)$-th layer then takes the input of the $\ell^*$-th layer as its own, i.e.,

$$\mathbf{X}^{(\ell^*+n)} = \mathbf{X}^{(\ell^*)} + f(\mathbf{X}^{(\ell^*)}, \theta^{(\ell^*+n)}). \tag{3}$$

In this work we adopt cosine similarity as the pruning metric for most cases, though our approach is agnostic to the choice of metric. Additional analyses of pruning metrics are provided in Appendix H.

However, we observe that *layer pruning introduces large mismatches in channel magnitudes at the pruning interface, which severely degrade model performance*, as shown in Figure 1. In Sections 3.2 and 3.3, we investigate two root causes of this issue and propose corresponding remedies.

### 3.2 Channel Magnitude Alignment

**Layer-wise Channel Mismatch.** We find that *mismatched channel magnitudes directly impair the performance of pruned LLMs*. As illustrated in Figure 1(a), hidden state magnitudes vary substantially across layers and channels. Removing layers exacerbates this discrepancy. Additional visualizations across different LLMs are provided in Appendix J.

To mitigate this, we statistically compute channel-wise scaling factors. For each channel $k$, we calculate the ratio of mean activation magnitudes between the inputs of the $(\ell^* + n)$-th and $\ell^*$-th layers over a calibration set, yielding a scaling vector $\mathbf{d} \in \mathbb{R}^C$ with entries

$$d_k(\ell^*, \ell^* + n) = \|\mathbf{X}_{:,k}^{(\ell^*+n)}\|_1 / \|\mathbf{X}_{:,k}^{(\ell^*)}\|_1. \tag{4}$$

**Quantitative Evaluation.** We further perturb $\mathbf{X}^{(\ell^*)}$ by a scaling factor $\alpha$ around $\mathbf{d}$, i.e.,

$$\mathbf{X}^{(\ell^*+n)} = \alpha\mathbf{X}^{(\ell^*)} \cdot \mathbf{d} + f(\alpha\mathbf{X}^{(\ell^*)} \cdot \mathbf{d}, \theta^{(\ell^*+n)}). \tag{5}$$

Figure 1(b) shows that $\mathbf{X}^{(\ell^*)} \cdot \mathbf{d}$ best restores magnitude alignment and improves perplexity. Deviating from the optimal scaling ($\alpha \neq 1$) results in substantial performance degradation.

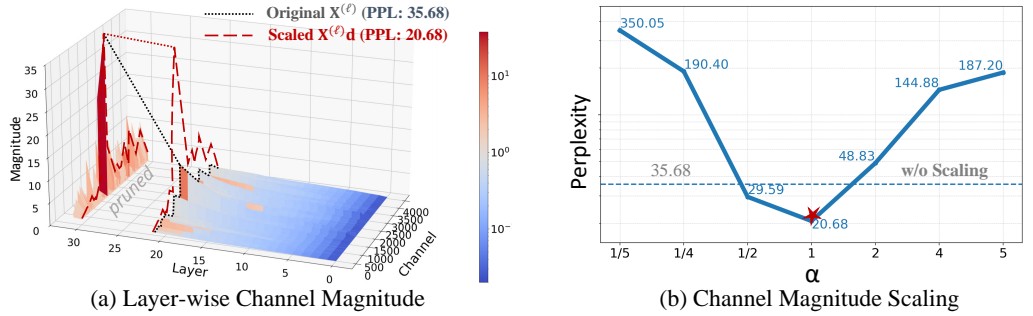

(a) Layer-wise Channel Magnitude  (b) Channel Magnitude Scaling

Figure 1: Visualization of layer-wise channel mismatch in pruned LLMs. Removing layers introduces magnitude mismatches, which we address using channel magnitude alignment.

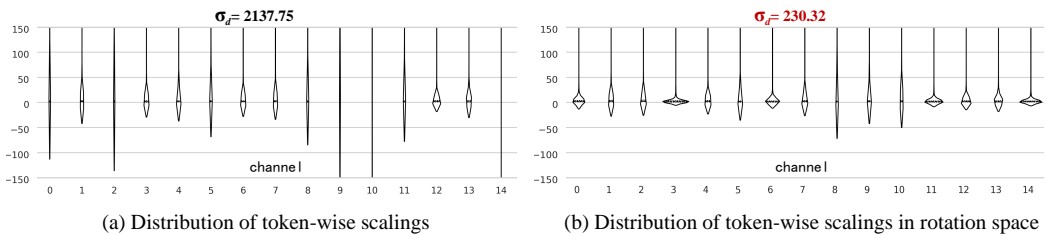

(a) Distribution of token-wise scalings  (b) Distribution of token-wise scalings in rotation space

Figure 2: Violin plot of token-wise scaling mismatch in pruned LLMs. The violin width represents the estimated probability density of token-wise scalings. After applying the Hadamard transformation, the scaling distributions become more concentrated, indicating reduced variance across tokens.

### 3.3 Token Magnitude Smoothing

**Token-wise Scaling Mismatch.** As suggested by recent research [32, 53], there are massive outliers specific to particular tokens (e.g., [BOS] and delimiter tokens) with magnitudes over $10^3$. Consequently, *a single channel scaling $d_k$ may fail to fit all tokens within that channel*, as illustrated in Figure 2(a). We quantify this mismatch by computing the standard deviation of token-wise scalings:

$$\sigma_{\mathbf{d}}(\ell^*, \ell^* + n) = \frac{1}{BC} \sum_{i=1}^{B} \sum_{k=1}^{C} \sigma \left( \frac{|\mathbf{X}_{i,k}^{(\ell^*+n)}|}{|\mathbf{X}_{i,k}^{(\ell^*)}|} \right), \quad (6)$$

where $\mathbf{X}_{i,k} \in \mathbb{R}^L$ denotes the activations of channel $k$ for batch $i$, and $\sigma(\cdot)$ is the standard deviation. A small $\sigma_{\mathbf{d}}$ indicates consistent scaling across tokens. However, we observe $\sigma_{\mathbf{d}} = 2137.75$ when pruning 9 layers from LLaMA-2-7B, reflecting severe token-level mismatch.

**Hadamard Transformation.** Recent works [30, 34, 4, 45] show that applying Hadamard transformations suppresses outliers. A Walsh–Hadamard matrix [23] of size $C = 2^n$ can be constructed recursively:

$$\begin{cases} \mathbf{H}_2 = \frac{1}{\sqrt{2}} \begin{bmatrix} 1 & 1 \\ 1 & -1 \end{bmatrix} \\ \mathbf{H}_{2^n} = \mathbf{H}_2 \otimes \mathbf{H}_{2^{n-1}} \end{cases}. \quad (7)$$

For $C \neq 2^n$, we follow [4] and factorize $C = 2^n m$ to construct $\mathbf{H}_C = \mathbf{H}_{2^n} \otimes \mathbf{H}_m$. The orthogonality of Hadamard matrix (i.e., $\mathbf{H}^\top \mathbf{H} = \mathbf{I}$) makes the following transformations equivalent:

$$\mathbf{X}^{(\ell^*)} = (\mathbf{X}^{(\ell^*)} \mathbf{H}) \mathbf{H}^\top, \quad (8)$$

where $\mathbf{X}^{(\ell^*)} \mathbf{H}$ denotes the rotated activations of $\ell^*$-th layer. The rotation effectively redistributes outliers among all channels and encourages a more balanced distribution of activation across channels. We also provide mathematical proofs in Appendix G. With the rotated activations, it is thus more friendly to share the same scaling parameter $\mathbf{d}$ for all tokens, with $\sigma_{\mathbf{d}}$ down to 230.32.

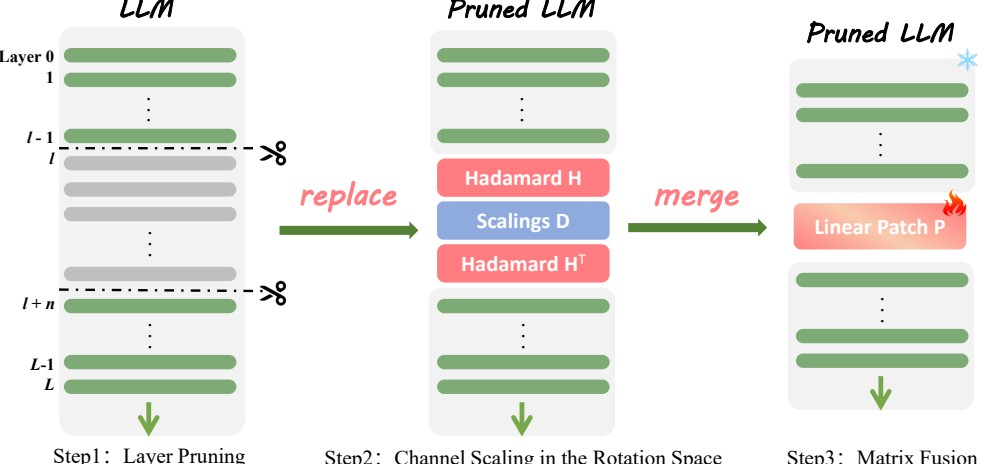

Step1: Layer Pruning    Step2: Channel Scaling in the Rotation Space    Step3: Matrix Fusion

Figure 3: Overview of LINEARPATCH. First, layers are pruned using a specified metric. Next, channel-wise scalings are estimated in the Hadamard-transformed space. Finally, the scalings are fused with Hadamard transformations to form LINEARPATCH, which supports efficient fine-tuning.

### 3.4   LINEARPATCH: the Ultimate Recipe

LINEARPATCH acts as a plug-and-play module that can be easily integrated to improve the layer-pruned LLMs. Specifically, we first apply Hadamard rotation to $\mathbf{X}^{(\ell^*)}$, then scale in the rotated space with $\mathbf{D} = \mathrm{diag}(\mathbf{d}) \in \mathbb{R}_+^{C \times C}$. The operations are fused into a single symmetric matrix $\mathbf{P} \in \mathbb{S}_{++}^n$:

$$\mathbf{X_{new}}^{(\ell^*)} = \mathbf{X}^{(\ell^*)}\mathbf{H}\mathbf{D}\mathbf{H}^\top = \mathbf{X}^{(\ell^*)}\mathbf{P}, \tag{9}$$

where the last equality stems from the spectral theorem [21], i.e., any real symmetric matrix can be decomposed into an orthogonal matrix of eigenvectors ($\mathbf{H}$) and a diagonal matrix of real eigenvalues ($\mathbf{D}$). The process is also visualized in Figure 3, where the patch matrix $\mathbf{P}$ can effectively bridge the gap for layer-pruned LLMs. In addition, LINEARPATCH also *reduces transformation overhead and enables efficient fine-tuning*, since only a single GEMM operation for the matrix multiplication is required, rather than three distinct GEMM operations.

**Memory-Efficient Offline Knowledge Distillation.**   Conventional knowledge distillation requires loading both the teacher and student models into GPU memory, which is prohibitive for LLMs due to the extreme memory overhead. In contrast, building on Equation (9), LINEARPATCH supports a memory-efficient offline distillation strategy. By storing only the inputs and outputs of the teacher model, we keep it offline during knowledge distillation.

Given a small training corpus $\mathbf{X} \in \mathcal{T}$ (e.g., 5,000 samples), we extract the top-$K$ output logits probability distribution $\mathbf{o}_t$ and their indices from the teacher's vocabulary. We set $K = 100$ in practice, which reduces memory usage by $320\times$ compared to storing the full 32K vocabulary. An ablation study on $K$ is provided in Appendix F.

Similarly, we collect the top-$K$ logits probability distribution $\mathbf{o}_s$ from the student model (the pruned LLM) using the same indices as the teacher. We then optimize the patch matrix $\mathbf{P}$ by minimizing the Kullback-Leibler (KL) divergence [9, 11, 12] between the two distributions:

$$\min_{\mathbf{P}} \; \mathbb{E}_{\mathbf{X} \in \mathcal{T}} \, \mathbf{KL}\left(\mathbf{o}_t, \mathbf{o}_s\right). \tag{10}$$

During fine-tuning, we remove the positive-definite constraint on $\mathbf{P}$ to allow greater flexibility, while freezing all other model parameters to minimize memory usage. The entire process is lightweight: for example, fine-tuning LLaMA-2-7B requires only 30 minutes on a single NVIDIA V100 GPU.

We also experimented with minimizing the mean squared error (MSE) between the pruned and teacher layers following [8], but found that MSE led to overfitting and consistently inferior results compared to KL-based logits distillation.

# 4 Experiments

## 4.1 Setup

**Models and Baselines.** We evaluate LINEARPATCH on several open-source LLMs, including LLaMA-2-7B/13B [47], LLaMA-3-8B [16], Baichuan2-7B [54], and DeepSeek-R1-Distill [19].

We compare against state-of-the-art layer pruning methods with diverse pruning metrics: gradient-based LLM-Pruner [35], perplexity-based SLEB [42], Taylor-based Shortened LLaMA [26], and cosine similarity-based ShortGPT [37] and LLM-Streamline [8]. For LLM-Pruner, we follow the block-wise pruning setup with the C4 calibration set using the official code. For SLEB and Shortened LLaMA, we adopt the official implementation or use the released pruned-layer indices.[2] For ShortGPT and LLM-Streamline, we reproduce the methods strictly following their published descriptions.s

**Evaluation.** We use three benchmarks for evaluation: perplexity (PPL), Massive Multitask Language Understanding (MMLU), and commonsense question answering (QA). For PPL, we evaluate language modeling on WikiText-2 (WIKI-2) [38], C4 [39], and PTB [36]. For MMLU, we report five-shot accuracy on the full benchmark [22]. For QA, we evaluate on nine commonsense QA tasks, reporting the average (Avg.), the weighted average (Weighted avg.) for MMLU, and both the average and retained performance (RP) for QA. Further details are provided in Appendix B.

## 4.2 Implementation Details

**Calibration and Fine-tuning.** To determine pruned layers and initialize channel-wise scaling parameters, we use 128 randomly sampled sentences with sequence length 2048 from WikiText-2 for calibration. Ablation studies on the calibration set size and distillation dataset size are reported in Appendix C, calibration dataset in Appendix D and Appendix E, respectively. For fine-tuning LINEARPATCH, we use AdamW with a learning rate of $1e-4$, training for one epoch on 5,000 WikiText-2 sentences of length 2048.

**Resource Consumption.** Our implementation is based on PyTorch. All experiments are conducted on a single NVIDIA V100 GPU with 24GB memory. For a 7B model, initialization of LINEARPATCH takes $\sim$30 seconds, while the optional fine-tuning process completes within 30 minutes.

**Pruning Configurations.** We follow prior work and restrict pruning ratios to below 30%. Detailed layer pruning configurations, including the officially released model links, the number and indices of pruned layers for each baseline, are listed in Appendix A.

## 4.3 Main Results

Our proposed LINEARPATCH can be seamlessly integrated with existing layer-wise pruning methods. Throughout the experiments, we denote the combination of LINEARPATCH with ShortGPT and LLM-Streamline as $\text{LINEARPATCH}_{[S]}$ and $\text{LINEARPATCH}_{[L]}$, respectively. In most cases, these two approaches prune similar sets of layers, therefore, we collectively refer to them as $\text{LINEARPATCH}_{[S/L]}$.

### 4.3.1 Comparison on Training-free Methods

We begin by evaluating the effectiveness of LINEARPATCH under training-free settings. Specifically, we focus on commonsense question answering (QA) benchmarks and perplexity (PPL) benchmarks, which are widely adopted to measure the retained reasoning and language modeling capabilities of pruned large language models (LLMs). To ensure fairness in comparison, none of the considered approaches are allowed to perform fine-tuning. In particular, for LLM-Pruner we exclude its LoRA-based fine-tuning stage, while for LLM-Streamline we follow its official protocol and denote the variant that discards layer replacement and offline distillation as LLM-Streamline (None). Results on more LLM backbones and additional benchmarks can be found in Appendix I.

---

[2]LLM-Pruner: `https://github.com/horseee/LLM-Pruner`; SLEB: `https://github.com/jiwonsong-dev/SLEB`; Shortened LLaMA: `https://github.com/Nota-NetsPresso/shortened-llm`.

Table 1: Comparison on QA benchmark with training-free methods. $L_p$ denotes the number of pruned layers and $L_t$ denotes the total number of layers of the model. The Ratio column represents the proportion (%) of pruning parameters to the total parameters of the model. Avg. column denotes the average accuracy (%) and RP column denotes the retained performance (%). Same interpretation is adopted in all the tables.

| Model | $L_p/L_t$ | Method | Ratio | ARC-c | ARC-e | BoolQ | HeSw | PIQA | WG | WSC | Race-h | CoPa | Avg. | RP |
|---|---|---|---|---|---|---|---|---|---|---|---|---|---|---|
| LLaMA-2-7B | 0/32 | Dense | - | 46.25 | 74.58 | 77.74 | 75.97 | 79.11 | 68.98 | 80.59 | 39.62 | 87.00 | 69.98 | 100 |
| | 9/32 | LLMPruner | 26.99 | 31.91 | 52.90 | 62.42 | 54.41 | 71.33 | 53.20 | 65.57 | 28.52 | 79.00 | 55.47 | 78.14 |
| | 9/32 | SLEB | 27.03 | 31.91 | 52.31 | 46.09 | 58.28 | 69.59 | 58.25 | 69.23 | 32.25 | 79.00 | 55.21 | 78.41 |
| | 9/32 | ShortGPT | 27.03 | 32.76 | 48.61 | 62.17 | 56.17 | 64.36 | 64.33 | 71.06 | 32.25 | 77.00 | 56.52 | 80.29 |
| | 9/32 | LLM-Streamline (None) | 27.03 | 32.76 | 48.61 | 62.17 | 56.17 | 64.36 | 64.33 | 71.06 | 32.25 | 77.00 | 56.52 | 80.29 |
| | 9/32 | LINEARPATCH$_{[S/L]}$ | 26.78 | 33.45 | 55.22 | 62.14 | 57.67 | 67.46 | 65.11 | 77.29 | 34.93 | 79.00 | **59.14** | **84.08** |
| | 7/32 | LLMPruner | 20.56 | 35.24 | 60.61 | 62.42 | 61.66 | 75.41 | 54.78 | 71.43 | 31.67 | 80.00 | 59.25 | 83.80 |
| | 7/32 | SLEB | 21.02 | 33.02 | 56.57 | 63.91 | 62.49 | 73.07 | 58.96 | 69.23 | 32.06 | 84.00 | 59.26 | 83.66 |
| | 7/32 | ShortGPT | 21.02 | 36.18 | 55.89 | 62.17 | 62.66 | 70.40 | 65.98 | 77.29 | 33.78 | 81.00 | 60.59 | 86.06 |
| | 7/32 | LLM-Streamline (None) | 21.02 | 36.18 | 55.89 | 62.17 | 62.66 | 70.40 | 65.98 | 77.29 | 33.78 | 81.00 | 60.59 | 86.06 |
| | 7/32 | LINEARPATCH$_{[S/L]}$ | 20.78 | 37.63 | 61.24 | 62.14 | 63.49 | 70.46 | 65.90 | 79.49 | 36.46 | 85.00 | **62.42** | **88.88** |
| LLaMA-2-13B | 0/40 | Dense | - | 49.06 | 77.40 | 80.61 | 79.35 | 80.52 | 72.38 | 86.81 | 40.48 | 91.00 | 73.07 | 100 |
| | 10/40 | LLMPruner | 23.90 | 39.51 | 67.26 | 65.84 | 72.24 | 77.91 | 57.30 | 73.26 | 33.59 | 85.00 | 63.55 | 86.32 |
| | 10/40 | SLEB | 24.37 | 39.93 | 66.04 | 66.76 | 68.24 | 75.63 | 63.61 | 75.46 | 36.94 | 84.00 | 64.07 | 87.54 |
| | 10/40 | ShortGPT | 24.37 | 43.00 | 63.51 | 58.20 | 69.29 | 72.52 | 69.85 | 81.32 | 36.84 | 87.00 | 64.61 | 88.45 |
| | 10/40 | LLM-Streamline (None) | 24.37 | 43.00 | 63.51 | 58.20 | 69.29 | 72.52 | 69.85 | 81.32 | 36.84 | 87.00 | 64.61 | 88.45 |
| | 10/40 | LINEARPATCH$_{[S/L]}$ | 24.17 | 44.20 | 65.53 | 62.39 | 70.15 | 73.83 | 69.61 | 81.68 | 38.09 | 89.00 | **66.05** | **90.49** |
| | 8/40 | LLMPruner | 19.48 | 41.98 | 67.51 | 63.33 | 68.76 | 76.50 | 56.51 | 68.86 | 32.06 | 85.00 | 62.28 | 84.78 |
| | 8/40 | SLEB | 19.50 | 36.43 | 61.83 | 62.32 | 67.03 | 75.08 | 62.51 | 78.02 | 34.83 | 83.00 | 62.34 | 84.74 |
| | 8/40 | ShortGPT | 19.50 | 44.03 | 67.38 | 57.00 | 72.38 | 75.24 | 69.61 | 79.85 | 38.18 | 89.00 | 65.85 | 90.27 |
| | 8/40 | LLM-Streamline (None) | 19.50 | 44.03 | 67.38 | 57.00 | 72.38 | 75.24 | 69.61 | 79.85 | 38.18 | 89.00 | 65.85 | 90.27 |
| | 8/40 | LINEARPATCH$_{[S/L]}$ | 19.30 | 44.54 | 69.53 | 67.74 | 73.02 | 75.68 | 69.38 | 82.78 | 39.71 | 92.00 | **68.26** | **93.45** |
| LLaMA-3-8B | 0/32 | Dense | - | 53.41 | 77.78 | 81.28 | 79.16 | 80.85 | 72.85 | 86.45 | 40.19 | 89.00 | 73.44 | 100 |
| | 7/32 | LLMPruner | 19.37 | 35.32 | 59.30 | 55.23 | 51.48 | 72.58 | 59.98 | 67.03 | 31.39 | 81.00 | 57.03 | 77.12 |
| | 7/32 | SLEB | 19.01 | 34.04 | 60.06 | 45.17 | 62.01 | 74.05 | 55.01 | 67.40 | 32.82 | 74.00 | 56.06 | 76.08 |
| | 7/32 | ShortGPT | 19.01 | 42.41 | 56.65 | 65.26 | 64.70 | 70.89 | 71.19 | 73.63 | 34.16 | 75.00 | 61.54 | 83.79 |
| | 7/32 | LLM-Streamline (None) | 19.01 | 28.92 | 39.56 | 38.07 | 33.26 | 59.47 | 55.56 | 59.71 | 24.02 | 60.00 | 44.29 | 59.99 |
| | 7/32 | LINEARPATCH$_{[S]}$ | 18.80 | 43.17 | 60.82 | 75.66 | 66.74 | 72.85 | 70.17 | 75.82 | 37.51 | 77.00 | **64.42** | **87.82** |
| | 7/32 | LINEARPATCH$_{[L]}$ | 18.80 | 34.39 | 51.26 | 57.52 | 49.31 | 63.33 | 63.22 | 72.53 | 29.95 | 67.00 | 54.28 | 73.57 |
| | 5/32 | LLMPruner | 13.39 | 39.51 | 68.10 | 71.28 | 64.69 | 76.33 | 64.48 | 74.36 | 35.60 | 78.00 | 63.59 | 86.23 |
| | 5/32 | SLEB | 13.58 | 39.68 | 66.16 | 54.71 | 67.39 | 75.90 | 62.51 | 73.63 | 34.16 | 83.00 | 61.90 | 83.88 |
| | 5/32 | ShortGPT | 13.58 | 45.56 | 63.51 | 73.12 | 70.13 | 74.92 | 71.19 | 75.09 | 36.94 | 79.00 | 65.50 | 89.27 |
| | 5/32 | LLM-Streamline (None) | 13.58 | 47.35 | 66.20 | 73.52 | 71.10 | 74.27 | 71.03 | 76.56 | 36.65 | 84.00 | 66.74 | 90.84 |
| | 5/32 | LINEARPATCH$_{[S]}$ | 13.37 | 45.73 | 68.60 | 73.30 | 70.71 | 76.01 | 73.09 | 79.85 | 38.18 | 82.00 | 67.50 | 91.91 |
| | 5/32 | LINEARPATCH$_{[L]}$ | 13.37 | 48.55 | 70.71 | 74.25 | 72.52 | 76.71 | 73.95 | 81.32 | 38.37 | 86.00 | **69.15** | **94.15** |

**Results on QA Benchmarks.** As presented in Table 1, LINEARPATCH consistently achieves improvements over prior training-free pruning methods across multiple backbones. For instance, on the LLaMA-2-7B model with 7 out of 32 layers pruned, LINEARPATCH attains a retained performance ratio of 88.88%, clearly outperforming LLM-Pruner (83.80%) and SLEB (83.66%). On the more recent LLaMA-3-8B model with 5 out of 32 layers pruned, our method yields an even more substantial improvement, achieving 94.15% retained performance, while LLM-Pruner and SLEB fall behind at 86.23% and 83.88%, respectively. When ShortGPT/LLM-Streamline is employed as the baseline pruning method, LINEARPATCH$_{[S/L]}$ achieves the state-of-the-art performance. For example, on LLaMA-2-7B with 9 out of 32 layers pruned, LINEARPATCH$_{[S/L]}$ reaches 84.08% retained performance, representing a 3.79% gain over both ShortGPT and LLM-Streamline. Similarly, on the larger LLaMA-13B model with 8 out of 40 layers pruned, LINEARPATCH$_{[S/L]}$ secures 93.45% retained performance, outperforming both baselines by 3.18%. These results highlight that LINEARPATCH provides consistent benefits when incorporated into existing pruning pipelines.

**Results on PPL Benchmarks.** We further examine the performance of pruned models on PPL benchmarks, which offer a direct measure of language modeling ability. Lower perplexity values correspond to stronger generative modeling performance. As shown in Table 2, SLEB equipped with a PPL-based pruning metric displays moderate advantages in this evaluation but struggles to maintain accuracy on QA tasks. By contrast, approaches using cosine similarity-based pruning are more balanced, and among them, LINEARPATCH consistently achieves the best overall performance. For example, on LLaMA-2-13B with 8 out of 40 layers pruned, LINEARPATCH achieves an average perplexity of 18.10, dramatically surpassing LLM-Pruner (35.06) and SLEB (36.61). A particularly striking case occurs on the LLaMA-3-8B model with 7 out of 32 layers pruned: LLM-Streamline nearly collapses, producing an unacceptably high average PPL of 2839.3, indicating severe failure in retaining generative capability. In sharp contrast, LINEARPATCH$_{[L]}$ effectively rescues the model without any additional training, restoring its performance to a functional and competitive level. This

Table 2: Comparison on PPL benchmark with training-free methods over LLaMA-2-7B and LLaMA-3-8B with 7 out of 32 layers pruned.

| Model | Method | WIKI-2 | C4 | PTB | PPL avg. |
|---|---|---|---|---|---|
| | Dense | 5.47 | 6.97 | 22.51 | 11.65 |
| | SLEB | 9.14 | 11.21 | 38.45 | 19.60 |
| | +LINEARPATCH | 8.77 | 10.66 | 38.30 | **19.24** |
| | Taylor+ | 18.45 | 20.99 | 62.18 | 33.87 |
| LLaMA-2-7B | +LINEARPATCH | 13.84 | 15.28 | 48.26 | **25.79** |
| | ShortGPT | 18.45 | 20.99 | 62.18 | 33.87 |
| | +LINEARPATCH | 13.22 | 14.58 | 45.97 | **24.59** |
| | LLM-Streamline (None) | 18.45 | 20.99 | 62.18 | 33.87 |
| | +LINEARPATCH | 13.22 | 14.58 | 45.97 | **24.59** |
| | Dense | 6.14 | 8.88 | 10.59 | 8.54 |
| | SLEB | 13.12 | 16.76 | 21.04 | 16.97 |
| | +LINEARPATCH | 11.97 | 15.74 | 19.55 | **15.75** |
| | Taylor+ | 2287.86 | 1491.38 | 4741.90 | 2840.38 |
| LLaMA-3-8B | +LINEARPATCH | 208.88 | 235.63 | 264.97 | **236.49** |
| | ShortGPT | 57.76 | 50.13 | 67.39 | 58.43 |
| | +LINEARPATCH | 25.67 | 28.38 | 31.22 | **28.42** |
| | LLM-Streamline (None) | 2287.73 | 1491.37 | 4738.81 | 2839.30 |
| | +LINEARPATCH | 69.82 | 96.68 | 88.79 | **85.10** |

Table 3: Comparison on QA benchmark with the SOTA post-training method LLM-Streamline (FFN).

| Model | $L_p/L_t$ | Method | ARC-c | ARC-e | BoolQ | HeSw | PIQA | WG | WSC | Race-h | CoPa | Avg. | RP |
|---|---|---|---|---|---|---|---|---|---|---|---|---|---|
| | 0/32 | Dense | 46.25 | 74.58 | 77.74 | 75.97 | 79.11 | 68.98 | 80.59 | 39.62 | 87.00 | 69.98 | 100 |
| LLaMa-2-7B | 7/32 | LLM-Streamline + FT | 38.23 | 60.48 | 70.18 | 63.75 | 69.86 | 67.48 | 80.95 | 37.51 | 79.00 | 63.05 | 90.00 |
| | 7/32 | LINEARPATCH$_{[L]}$ | 37.63 | 61.24 | 62.14 | 63.49 | 70.46 | 65.90 | 79.49 | 36.46 | 85.00 | 62.42 | 88.88 |
| | 7/32 | LINEARPATCH$_{[L]}$ + FT | 38.23 | 64.35 | 65.32 | 69.33 | 73.23 | 67.40 | 83.88 | 38.37 | 87.00 | **65.23** | 92.83 |
| | 0/32 | Dense | 53.41 | 77.78 | 81.28 | 79.16 | 80.85 | 72.85 | 86.45 | 40.19 | 89.00 | 73.44 | 100 |
| LLaMA-3-8B | 5/32 | LLM-Streamlines + FT | 30.03 | 39.94 | 45.09 | 49.19 | 59.79 | 67.80 | 81.32 | 31.39 | 71.00 | 55.09 | 74.34 |
| | 5/32 | LINEARPATCH$_{[L]}$ | 48.55 | 70.71 | 74.25 | 72.52 | 76.71 | 73.95 | 81.32 | 38.37 | 86.00 | 69.15 | 94.15 |
| | 5/32 | LINEARPATCH$_{[L]}$ + FT | 48.12 | 72.77 | 70.98 | 74.63 | 77.42 | 74.03 | 84.62 | 38.56 | 89.00 | **70.01** | 95.16 |

demonstrates the robustness of LINEARPATCH in mitigating instability issues inherent in pruning strategies, ensuring stable recovery of language modeling performance.

**Results on PPL Benchmarks.** We further assess the effect of pruning on PPL benchmarks, which provide a direct evaluation of the language modeling ability of pruned LLMs. Lower perplexity values indicate stronger generative capacity. As summarized in Table 2, LINEARPATCH consistently improves the robustness and performance of diverse pruning baselines across both LLaMA-2-7B and LLaMA-3-8B models. On LLaMA-2-7B with 7 out of 32 layers pruned, LINEARPATCH yields significant improvements for all baseline methods. For instance, when combined with ShortGPT and LLM-Streamline (None), LINEARPATCH decreases the average PPL from 33.87 to 24.59, highlighting its ability to consistently mitigate the performance degradation caused by pruning. On LLaMA-3-8B with 7 out of 32 layers pruned, the advantage of LINEARPATCH is even more pronounced. When applied to SLEB, LINEARPATCH improves average PPL from 16.97 to 15.75; for Taylor+, it reduces catastrophic degradation (2840.38) to a reasonable 236.49; and for ShortGPT, it lowers average PPL from 58.43 to 28.42. These results clearly demonstrate that LINEARPATCH serves as a general and effective enhancement across different pruning strategies.

### 4.3.2 Comparison on Post-training Methods

Beyond training-free pruning, we next compare LINEARPATCH with the state-of-the-art post-training method LLM-Streamline, under identical fine-tuning configurations (i.e., same dataset, sample count, and learning rate). This setup allows us to fairly isolate the benefit introduced by our lightweight LINEARPATCH mechanism.

**Results on QA Benchmarks.** Table 3 illustrates that LINEARPATCH provides clear advantages when combined with post-training fine-tuning. On LLaMA-2-7B with 7 out of 32 layers pruned, LINEARPATCH$_{[L]}$+FT achieves 65.23% accuracy, outperforming LLM-Streamline (63.05%). Similarly, on LLaMA-3-8B with 5 out of 32 layers pruned, our method achieves 70.01% accuracy, whereas LLM-Streamline fails to deliver competitive results due to its reliance on randomly ini-

Table 4: Comparison on PPL benchmark with SOTA post-training method LLM-Streamline.

| Model | $L_p/L_t$ | Method | Ratio | WIKI-2 | C4 | PTB | Avg. |
|---|---|---|---|---|---|---|---|
| LLaMA-2-7B | 0/32 | Dense | 0 | 5.47 | 6.97 | 22.51 | 11.65 |
| | 7/32 | LLM-Streamline (FFN) + FT | 19.01 | 9.60 | 17.10 | 47.04 | 24.58 |
| | 7/32 | LINEARPATCH$_{[L]}$ | 20.78 | 13.22 | 14.58 | 45.97 | 24.59 |
| | 7/32 | LINEARPATCH$_{[L]}$ + FT | 20.78 | 8.09 | 11.25 | 32.48 | **17.27** |
| LLaMA-3-8B | 0/32 | Dense | - | 6.14 | 8.88 | 10.59 | 8.54 |
| | 5/32 | LLM-Streamline (FFN) + FT | 11.39 | 383.15 | 201.60 | 101.35 | 228.70 |
| | 5/32 | LINEARPATCH$_{[L]}$ | 13.37 | 15.13 | 17.41 | 19.30 | 17.28 |
| | 5/32 | LINEARPATCH$_{[L]}$ + FT | 13.37 | 9.00 | 13.34 | 14.34 | **12.23** |

Table 5: Comparisons on tunable parameters, distillation type and loss functions. FT denotes fine-tuning. When distilling feature, we optimize the parameters of the replaced layer by aligning its output with MSE loss as LLM-Streamline. When distilling logits, we optimize the parameters of the replaced layer by aligning the model output logits with KL loss.

| Model | Parameters | Distillation | Loss | WIKI-2 | C4 | PTB | PPL Avg. | QA Avg. | QA RP |
|---|---|---|---|---|---|---|---|---|---|
| Dense | - | - | - | 5.47 | 6.97 | 22.51 | 11.65 | 69.98 | 100 |
| LLM-Streamline (FFN) + FT | FFN | Feature | MSE | 13.00 | 27.22 | 68.97 | 36.40 | 59.44 | 84.74 |
| | FFN | Logits | KL | - | - | - | - | - | NaN |
| LINEARPATCH$_{[L]}$+ FT | Symmetric matrix | Feature | MSE | 15.26 | 19.54 | 54.33 | 29.71 | 59.58 | 84.80 |
| | Diagonal matrix | Logits | KL | 17.51 | 18.64 | 51.64 | 29.26 | 59.40 | 84.45 |
| | Symmetric matrix | Logits | KL | **8.60** | **12.98** | 37.16 | **19.58** | **61.71** | **88.15** |

tialized replacement layers. Notably, the retained performance (RP) of LINEARPATCH$_{[L]}$+FT on LLaMA-3-8B reaches 95.16%, underscoring its ability to maintain nearly all original model accuracy despite aggressive layer pruning. These findings reinforce that LINEARPATCH is well suited for post-training recovery, offering both stability and efficiency advantages.

**Results on PPL Benchmarks.** The superiority of LINEARPATCH is also evident in PPL evaluations. As reported in Table 4, LINEARPATCH$_{[L]}$+FT consistently matches or exceeds LLM-Streamline across different pruning configurations. For instance, on LLaMA-2-7B with 7 out of 32 layers pruned, our method achieves an average PPL of 17.27 compared to 24.58 for LLM-Streamline. Likewise, on LLaMA-3-8B with 5 pruned layers, LINEARPATCH$_{[L]}$+FT attains 12.23 perplexity, reflecting its robustness and effectiveness in preserving language modeling quality under post-training scenarios.

### 4.4 Discussions and Ablation Studies

**Tunable Parameters and Loss Functions.** To further understand the design choices, we conducted comprehensive experiments on LLaMA-2-7B with 9 out of 32 layers pruned, systematically varying training settings such as loss functions and distillation targets (see Table 5). Results reveal that training configurations have a marked impact on performance. In particular, using model logits as the distillation target combined with KL divergence as the loss function yields the best results: an average PPL of 19.58 on language modeling tasks and an average QA accuracy of 61.71%, representing a 2.13% improvement over LLM-Streamline. Conversely, using replaced layer outputs for distillation tends to cause overfitting, resulting in inferior performance. Additionally, LLM-Streamline's strategy of employing a randomly initialized lightweight network introduces instability during training and can lead to degraded outcomes. These observations highlight the advantage of LINEARPATCH, which offers a principled and stable initialization, enabling more reliable performance recovery through lightweight fine-tuning.

**The Ingredients of LINEARPATCH.** We further ablate the contribution of individual components of LINEARPATCH, as reported in Table 6. Without any additional mechanism, a pruned model suffers severe degradation, with average PPL increasing to 56.10 and QA retained performance dropping to 80.29%. Introducing scaling parameters **d** already mitigates the degradation, improving PPL to 33.70 and QA retention to 83.56%. Incorporating the Hadamard rotation to construct the full LINEARPATCH **P** further enhances results, reducing PPL to 30.29 and pushing QA retention to 84.08%, corresponding to a relative improvement of 3.8% over baseline pruning. Finally, applying fine-tuning on top of LINEARPATCH provides additional substantial gains, with RP improved by

Table 6: Ablation study on the ingredients of LINEARPATCH over LLaMA-2-7B with 9 out of 32 layers pruned. $+\mathbf{d}$ applies channel scaling, $+\mathbf{P}$ (i.e., $\mathbf{HDH}^\top$) refers to the LINEARPATCH, and $+$FT denotes fine-tuning with knowledge distillation. Note that we omit ablating $\mathbf{H}$ since Hadamard transformation alone is an equivalent operation.

| | WIKI-2 | C4 | PTB | Avg. | ARC-c | ARC-e | BoolQ | HeSw | PIQA | WG | WSC | Race-h | CoPa | Avg. | RP |
|---|---|---|---|---|---|---|---|---|---|---|---|---|---|---|---|
| Dense | 5.47 | 6.97 | 22.51 | 11.65 | 46.25 | 74.58 | 77.74 | 75.97 | 79.11 | 68.98 | 80.59 | 39.62 | 87.00 | 69.98 | 100 |
| Vanilla | 35.68 | 36.10 | 96.52 | 56.10 | 32.76 | 48.61 | 62.17 | 56.17 | 64.36 | 64.33 | 71.06 | 32.25 | 77.00 | 56.52 | 80.29 |
| $+\mathbf{d}$ | 20.68 | 22.75 | 57.67 | 33.70 | 35.07 | 54.97 | 62.17 | 56.93 | 66.76 | 63.77 | 75.09 | 34.83 | 78.00 | 58.62 | 83.56 |
| $+\mathbf{P}$ | 18.60 | 19.28 | 53.00 | 30.29 | 33.53 | 55.22 | 62.14 | 57.69 | 67.41 | 65.04 | 77.29 | 34.93 | 79.00 | 59.14 | 84.08 |
| $+$ FT | 8.60 | 12.98 | 37.16 | **19.58** | 34.81 | 60.65 | 62.48 | 64.52 | 70.29 | 66.69 | 76.92 | 39.04 | 80.00 | **61.71** | **88.15** |

4.06%. These ablations confirm that each design element contributes meaningfully to the strong overall performance of our approach.

**Online Inference Overhead.** We also evaluate the online inference overhead of LINEARPATCH. Experiments are conducted using batch size 16, sequence length 2048, 1000 iterations, hidden size 4096 (aligned with LLaMA-2-7B), and float16 precision. Our method, implemented as a single linear layer inserted at the pruning interface, introduces virtually no latency in end-to-end inference compared to baseline methods. Empirical results indicate no measurable overhead relative to LLM-Streamline with its feed-forward network (FFN). Furthermore, LINEARPATCH is approximately $8\times$ smaller in parameter size and around $190\times$ faster than LLM-Streamline(FFN), making it significantly more efficient for deployment.

**Offline Storage Overhead.** Finally, we examine the offline storage overhead. With Top-K logits distillation, LINEARPATCH reduces storage requirements by up to $40\times$ for hidden size 4096, compared with LLM-Streamline during offline fine-tuning. This efficiency stems from the much smaller intermediate tensors that need to be stored by our approach. Such a reduction not only improves practical usability but also substantially lowers the resource cost of post-training adaptation, which is critical in large-scale deployment scenarios.

## 5 Conclusion

In this work, we introduce LINEARPATCH, a simple yet effective plug-and-play technique that addresses the critical issue of activation magnitude mismatch in layer-pruned large language models (LLMs). By leveraging the Hadamard transformation and channel-wise scaling, LINEARPATCH efficiently aligns activations across layers, significantly enhancing model performance with negligible inference overhead. Extensive empirical evaluations are conducted to demonstrate the effectiveness of the proposed approach. We hope the proposed LINEARPATCH can shed more light on simple and lightweight algorithms of LLM compression without compromising the performance.

## 6 Limitation and Broader Impact

**Limitation.** Layer pruning may unevenly degrade model performance across different tasks. For instance, while some question answering tasks might remain robust, complex reasoning or context-dependent tasks could suffer. Future work should establish a framework to evaluate trade-offs between efficiency gains and task-specific performance.

**Broader Impact.** Layer pruning methods significantly reduce the computational costs of deploying large language models, making them more accessible to a broader range of users. However, these methods do not address the social biases embedded in LLMs, which often stem from the training data and can affect fairness and inclusivity. It is crucial to ensure ethical deployment of LLMs.

## 7 Acknowledgment

This work was supported by the National Key R&D Program of China (2022YFB4701400/4701402), SSTIC Grant (KJZD20230923115106012, KJZD20230923114916032, GJHZ20240218113604008).

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

# A    Details on Pruned Models

For our experiments, we adopt the officially released LLMs from the sources listed in Table 7. The corresponding pruned layer indices for different configurations and models are comprehensively presented in Table 8.

Table 7: Download links to officially released LLMs.

| Model | Download Link |
|---|---|
| LLaMA-2-7B | https://huggingface.co/meta-llama/Llama-2-7B |
| LLaMA-2-13B | https://huggingface.co/meta-llama/Llama-2-13B |
| LLaMA-3-8B | https://huggingface.co/meta-llama/Llama-3.1-8B |
| DeepSeek-R1-Distill-Qwen-7B | https://huggingface.co/deepseek-ai/DeepSeek-R1-Distill-Qwen-7B |
| DeepSeek-R1-Distill-Llama-8B | https://huggingface.co/deepseek-ai/DeepSeek-R1-Distill-Llama-8B |
| Baichuan2-7B | https://huggingface.co/baichuan-inc/Baichuan2-7B-Base |

Table 8: Details of pruning settings.

| Model | $L_p/L_t$ | Method | Pruned layer index |
|---|---|---|---|
| LLaMA-2-7B | 9/32 | LINEARPATCH$_{[S/L]}$ | [21,30) |
| | 7/32 | LINEARPATCH$_{[S/L]}$ | [23,30) |
| | 9/32 | Taylor+ | [28,27,25,29,26,24,23,21,22] |
| | 7/32 | Taylor+ | [28,27,25,29,26,24,23] |
| | 9/32 | SLEB | [14,23,11,24,10,27,15,21,25] |
| | 7/32 | SLEB | [14,23,11,24,10,27,15] |
| LLaMA-3-8B | 7/32 | LINEARPATCH$_{[S]}$ | [22,29) |
| | 5/32 | LINEARPATCH$_{[S]}$ | [24,29) |
| | 7/32 | LINEARPATCH$_{[L]}$ | [23,30) |
| | 5/32 | LINEARPATCH$_{[L]}$ | [23,28) |
| | 7/32 | Taylor+ | [24,25,26,28,29,23,27] |
| | 5/32 | Taylor+ | [24,25,26,28,29] |
| | 7/32 | SLEB | [10,25,11,26,12,9,23] |
| | 5/32 | SLEB | [10,25,11,26,12] |
| LLaMA-2-13B | 10/40 | LINEARPATCH$_{[S/L]}$ | [26,36) |
| | 8/40 | LINEARPATCH$_{[S/L]}$ | [28,36) |
| Baichuan2-7B | 9/32 | LINEARPATCH$_{[S/L]}$ | [22,31) |
| | 7/32 | LINEARPATCH$_{[S/L]}$ | [23,30) |

# B    Details of Evaluation Benchmarks

We use a variety of benchmarks for model evaluation, including the perplexity (PPL) benchmark (measured by the average perplexity score), the Massive Multitask Language Understanding (MMLU) benchmark and the question answering (QA) benchmark for model evaluation.

**PPL.**    For PPL benchmarks, we report the perplexity of language generation on WikiText2 [38], C4 [39], and PTB [36] datasets.

**MMLU.**    For MMLU benchmark, we test the five-shot performance on the Massively Multitask Language Understanding (MMLU) datasets [22].

**Commonsense QA.**    For QA benchmark, we evaluate methods on 9 commonsense QA tasks: ARC-Challenge (ARC-c), ARC-Easy (ARC-e) [15], BoolQ [14], HellaSwag (HeSw) [56], PIQA [6], WinoGrande (WG) [1], WSC273 (WSC) [28], Race-high (Race-h) [27] and CoPA [41].

For MMLU benchmark, we use the official code. For PPL and QA benchmarks, we use the lm_eval library (version 0.4.4) from https://github.com/EleutherAI/lm-evaluation-harness.

## C   Ablation on Size of Calibration Set

We vary the size of the calibration set (a subset of WIKI-2) to evaluate its impact on the performance of LINEARPATCH in Table 9. The results show that a larger calibration set leads to better scaling parameters and improved performance, but the gains diminish beyond a certain size. A calibration set of 128 samples provides a good balance between computational efficiency and performance.

Table 9: Ablation on the number of calibration samples for scaling parameters statistics over LLaMA-2-7B with 9 out of 32 layers pruned.

| Num of samples | WIKI-2 | C4 | PTB |
|---|---|---|---|
| 64 | 18.61 | 19.29 | 53.03 |
| **128** | 18.60 | 19.28 | 53.00 |
| 256 | 18.60 | 19.28 | 53.00 |
| 512 | 18.61 | 19.28 | 53.01 |

## D   Ablation on Calibration Data

To thoroughly investigate the impact of different calibration datasets and validate the robustness of our method, we provided experimental results of different calibration sets in Table 10. The key findings are as follows:

- **Domain-Specific Calibration Boosts Performance**: When PTB is used as the calibration set, the perplexity (PPL) on PTB drops to 52.25, significantly outperforming other calibration sets. This indicates that domain-specific calibration enhances performance in that domain.

- **Stability Under Domain Mismatch**: It is common that using domain-matched calibration data yields the best results (diagonal entries in Table 10). We note that even with domain-mismatched calibration data, the increase in PPL is marginal. For example, calibrating with WIKI-2 results in the PPL of 19.28 on C4, only 0.03 higher than the domain-matched case (19.25). This confirms the stability of our method.

- **Robustness to Data Quality**: Despite C4 being the lowest-quality calibration set, the average PPL (30.52) is only slightly higher than the best result (30.29). This minimal degradation demonstrates the robustness of our method to calibration data quality.

Table 10: Ablation on calibration datasets over LLaMA-2-7B with 9 out of 32 layers pruned.

| Calibration set | Data quality | WIKI-2 | PTB | C4 | PPL avg. |
|---|---|---|---|---|---|
| WIKI-2 | best | **18.60** | 53.00 | 19.28 | **30.29** |
| PTB | mediate | 19.31 | **52.24** | 19.66 | 30.40 |
| C4 | poor | 19.11 | 53.21 | **19.25** | 30.52 |

## E   Ablation on Size of Distillation Dataset

To provide deeper analyses on how dataset size impacts recovery during knowledge distillation, we ablated the dataset size in the setting of removing 7 layers of LLaMA-2-7B. As shown in Table 11, larger dataset for knowledge distillation tends to produce better performance, but the benefits are limited above 5000 samples. The adopted setting weighs the performance and efficiency.

Table 11: Ablation study on the size of distillation dataset.

| Size | Cache | WIKI-2 | C4 | PTB | Avg. | ARC-c | ARC-e | BoolQ | HeSw | PIQA | WG | WSC | Race-h | CoPa | Avg. (%) |
|---|---|---|---|---|---|---|---|---|---|---|---|---|---|---|---|
| w/o FT | - | 13.22 | 14.58 | 45.97 | 24.59 | 37.63 | 61.24 | 62.14 | 63.49 | 70.46 | 65.90 | 79.49 | 36.46 | 85.00 | 59.60 |
| 2500 | 5G | 8.22 | 11.25 | 32.79 | 17.42 | 38.82 | 63.89 | 64.86 | 69.12 | 73.01 | 68.43 | 81.32 | 37.89 | 68.00 | 62.82 |
| 5000 | 10G | 8.09 | 11.25 | 32.48 | 17.27 | 38.23 | 64.35 | 65.32 | 69.33 | 73.23 | 67.40 | 83.88 | 38.37 | 87.00 | 65.23 |
| 7500 | 15G | 8.07 | 11.24 | 33.37 | 17.56 | 38.91 | 64.14 | 66.06 | 69.49 | 73.45 | 68.03 | 80.95 | 38.09 | 88.00 | 65.24 |

# F Ablation on $K$ in Top-K Logits for Knowledge Distillation

We ablate $K$={50, 100, 200} in top-K logits to show how $K$ influence on knowledge distillation for pruned model. We removed 7 layers on LLaMA-2-7B and performed ablation experiments according to the fine-tuning settings. As shown in Table 12, benefits and costs are well balanced when $K$=100.

Table 12: Ablation study on on $K$ in top-K logits for knowledge distillation.

| K | Cache | WIKI-2 | C4 | PTB | Avg. | ARC-c | ARC-e | BoolQ | HeSw | PIQA | WG | WSC | Race-h | CoPa | Avg. (%) |
|---|---|---|---|---|---|---|---|---|---|---|---|---|---|---|---|
| w/o FT | - | 13.22 | 14.58 | 45.97 | 24.59 | 37.63 | 61.24 | 62.14 | 63.49 | 70.46 | 65.90 | 79.49 | 36.46 | 85.00 | 59.60 |
| 50 | 5G | 8.73 | 11.63 | 34.68 | 18.35 | 38.14 | 63.30 | 67.37 | 69.90 | 72.36 | 68.19 | 80.95 | 38.47 | 88.00 | 65.19 |
| 100 | 10G | 8.09 | 11.25 | 32.48 | 17.27 | 38.23 | 64.35 | 65.32 | 69.33 | 73.23 | 67.40 | 83.88 | 38.37 | 87.00 | 65.23 |
| 200 | 20G | 7.78 | 11.03 | 31.51 | 16.77 | 38.05 | 64.73 | 64.71 | 69.02 | 73.07 | 67.56 | 82.78 | 37.42 | 87.00 | 64.93 |

# G Mathematical Proof on Outlier Reduction of Hadamard Transformation

We introduce incoherence processing [4, 7] into activation with Hadamard transformation. A activation matrix $\mathbf{X} \in \mathbb{R}^{n \times m}$ is $\mu$-incoherent if all $i$ and $j$, $|\mathbf{X}_{ij}| = |e_i^T \mathbf{X} e_j| \leq \mu/||\mathbf{X}||_F \sqrt{nm}$. Then, we obtain:

$$max(\mathbf{X}) \leq \mu ||\mathbf{X}||_F / \sqrt{nm}, \tag{11}$$

where $max$ is the element-wise max of the matrix, and $mn$ is the number of elements.

# H Results on Combination LINEARPATCH with More Layer Pruning Metric

LINEARPATCH allows for the combination of non-contiguous layer pruning settings, gradient-based and PPL-based layer pruning metric, combined with any pruning metric results in the improvement of performance. We demonstrate these in the following experiments.

## H.1 Combination with Non-contiguous Layer Pruning Metric

We integrated LINEARPATCH with non-contiguous layer pruning settings. ShortGPT [37] accesses the importance of each layers with the prevalent cosine similarity. We replaced each removed layer in ShortGPT with a LINEARPATCH. As shown in Table 13, combining LINEARPATCH with non-contiguous layer pruning still significantly improves the performance of the pruned LLM. For example, on the LLaMA-3-8B with 7 layers pruned, LINEARPATCH achieved a 3.51% improvement in performance on the QA task benchmark compared to ShortGPT.

## H.2 Combination with Gradient-based Layer Pruning Metric

We integrated LINEARPATCH with gradient-based layer pruning metric taylor+ [26]. We identify the redundant layers with gradient information and replaced each redundant layer with a LINEARPATCH. As shown in Table 14, combining LINEARPATCH with gradient-based layer pruning metric significantly improves the performance. For example, on the LLaMA-3-8B model with 7 layers pruned, LINEARPATCH achieved a 7.64% improvement in performance on the QA task benchmark, with average PPL reducing from 2840.38 to 236.49.

## H.3 Combination with PPL-based Layer Pruning Metric

We combined LINEARPATCH with the SLEB [42], which uses PPL-based pruning metric. SLEB selects layers with the least performance drop based on the PPL metric, and the removed layers are usually non-contiguous. We replaced each removed layer in SLEB with a single LINEARPATCH. As shown in Table 15, LINEARPATCH consistently improves the pruned model's performance when combined with PPL-based pruning metrics, without any fine-tuning.

Table 13: Combination with non-contiguous cosine similarity-based pruning metric.

| Model | $L_p/L_t$ | Method | WIKI-2 | C4 | PTB | AVG | ARC-c | ARC-e | BoolQ | HeSw | PIQA | WG | WSC | Race-h | CoPa | Avg. | RP |
|---|---|---|---|---|---|---|---|---|---|---|---|---|---|---|---|---|---|
| | 0/32 | Dense | 5.47 | 6.97 | 22.51 | 11.65 | 46.25 | 74.58 | 77.74 | 75.97 | 79.11 | 68.98 | 80.59 | 39.62 | 87.00 | 69.98 | 100 |
| LLaMA-2-7B | 7/32 | ShortGPT | 18.45 | 20.99 | 62.18 | 33.87 | 36.18 | 55.89 | 62.17 | 62.66 | 70.40 | 65.98 | 77.29 | 33.78 | 81.00 | 60.59 | 86.06 |
| | 7/32 | +LINEARPATCH | 13.84 | 15.28 | 48.26 | **25.79** | 37.71 | 60.14 | 62.17 | 63.32 | 70.78 | 65.59 | 78.75 | 35.98 | 84.00 | 62.05 | **88.35** |
| | 0/32 | Dense | 6.14 | 8.88 | 10.59 | 8.54 | 53.41 | 77.78 | 81.28 | 79.16 | 80.85 | 72.85 | 86.45 | 40.19 | 89.00 | 73.44 | 100 |
| LLaMA-3-8B | 7/32 | ShortGPT | 57.76 | 50.13 | 67.39 | 58.43 | 42.41 | 56.65 | 65.26 | 64.70 | 70.89 | 71.19 | 73.63 | 34.16 | 75.00 | 61.54 | 83.79 |
| | 7/32 | +LINEARPATCH | 34.77 | 33.45 | 42.38 | **36.87** | 41.98 | 60.06 | 76.33 | 66.46 | 72.63 | 70.09 | 75.46 | 35.69 | 80.00 | 64.30 | **87.30** |

Table 14: Combination with gradient-based layer pruning metric.

| Model | $L_p/L_t$ | Method | WIKI-2 | C4 | PTB | AVG | ARC-c | ARC-e | BoolQ | HeSw | PIQA | WG | WSC | Race-h | CoPa | Avg. | RP |
|---|---|---|---|---|---|---|---|---|---|---|---|---|---|---|---|---|---|
| | 0/32 | Dense | 5.47 | 6.97 | 22.51 | 11.65 | 46.25 | 74.58 | 77.74 | 75.97 | 79.11 | 68.98 | 80.59 | 39.62 | 87.00 | 69.98 | 100 |
| LLaMA-2-7B | 7/32 | Taylor+ | 18.45 | 20.99 | 62.18 | 33.87 | 36.18 | 55.89 | 62.17 | 62.66 | 70.40 | 65.98 | 77.29 | 33.78 | 81.00 | 60.59 | 86.06 |
| | 7/33 | +LINEARPATCH | 13.84 | 15.28 | 48.26 | **25.79** | 37.71 | 60.14 | 62.17 | 63.32 | 70.78 | 65.59 | 78.75 | 35.98 | 84.00 | 62.05 | **88.35** |
| | 0/32 | Dense | 6.14 | 8.88 | 10.59 | 8.54 | 53.41 | 77.78 | 81.28 | 79.16 | 80.85 | 72.85 | 86.45 | 40.19 | 89.00 | 73.44 | 100 |
| LLaMA-3-8B | 7/32 | Taylor+ | 2287.86 | 1491.38 | 4741.90 | 2840.38 | 29.01 | 39.56 | 38.04 | 33.24 | 59.30 | 55.49 | 59.71 | 24.02 | 60.00 | 44.26 | 59.97 |
| | 7/32 | +LINEARPATCH | 208.88 | 235.63 | 264.97 | **236.49** | 31.66 | 45.20 | 48.07 | 43.13 | 63.87 | 57.54 | 63.37 | 27.85 | 67.00 | 49.74 | **67.43** |

Table 15: Combination with PPL-based layer pruning metric.

| Model | $L_p/L_t$ | Method | WIKI-2 | C4 | PTB | AVG | ARC-c | ARC-e | BoolQ | HeSw | PIQA | WG | WSC | Race-h | CoPa | Avg. | RP |
|---|---|---|---|---|---|---|---|---|---|---|---|---|---|---|---|---|---|
| | 0/32 | Dense | 5.47 | 6.97 | 22.51 | 11.65 | 46.25 | 74.58 | 77.74 | 75.97 | 79.11 | 68.98 | 80.59 | 39.62 | 87.00 | 69.98 | 100 |
| LLaMA-2-7B | 7/32 | SLEB | 9.14 | 11.21 | 38.45 | 19.60 | 33.02 | 56.57 | 63.91 | 62.49 | 73.07 | 58.96 | 69.23 | 32.06 | 84.00 | 59.26 | 83.66 |
| | 7/32 | +LINEARPATCH | 8.77 | 10.66 | 38.30 | **19.24** | 32.17 | 57.53 | 66.39 | 61.73 | 73.29 | 59.43 | 69.69 | 32.63 | 83.00 | 59.54 | **84.04** |
| | 0/32 | Dense | 6.14 | 8.88 | 10.59 | 8.54 | 53.41 | 77.78 | 81.28 | 79.16 | 80.85 | 72.85 | 86.45 | 40.19 | 89.00 | 73.44 | 100 |
| LLaMA-3-8B | 7/32 | SLEB | 13.12 | 16.76 | 21.04 | 16.97 | 34.04 | 60.06 | 45.17 | 62.01 | 74.05 | 55.01 | 67.40 | 32.82 | 74.00 | 56.06 | 76.08 |
| | 7/32 | +LINEARPATCH | 11.97 | 15.74 | 19.55 | **15.75** | 33.79 | 61.83 | 48.53 | 61.95 | 74.37 | 55.49 | 65.57 | 33.49 | 78.00 | 57.00 | **77.30** |

# I  Results on More Models and Benchmarks

## I.1  Comparison on QA Benchmarks with Training-free Methods

Table 16 shows more results on QA benchmarks. On the Baichuan2-7B model with 7 out of 32 layers pruned, LINEARPATCH$_{[S/L]}$ achieves a retained performance ratio of 87.27%, leading both ShortGPT and LLM-Streamline with 3.39%. When it comes to 9 out of 32 layers pruned, LINEARPATCH$_{[S/L]}$ still maintains the 81.66% of the original performance, outperforming ShortGPT and LLM-Streamline by 4.56%. We also provide the results of the latest DeepSeek-R1-Distill-Llama-8B and DeepSeek-R1-Distill-Qwen-7B models. The results show that LINEARPATCH$_{[L]}$ is applicable for latest GQA-based large language models.

## I.2  Comparison on MMLU Benchmarks with Training-free Methods

We evaluate the proposed LINEARPATCH method on the MMLU tasks across multiple models in Table 17. Overall, LINEARPATCH demonstrates significant improvements in weighted average accuracy across different models and pruning ratios. For example, it attains weighted average accuracy of 63.84% for LLaMA-3-8B with 5 out of 32 layers pruned, outperforming the best results from other methods. Similarly, on LLaMA-2-13B, it reaches 53.96% and 54.01% for 10 and 8 out of 40 layers pruned, respectively, where SLEB almost collapsed in the same case. These results highlight the robustness and effectiveness of LINEARPATCH in enhancing the performance of layer-pruned large language models on MMLU tasks, demonstrating its potential as a simple yet powerful solution for reviving pruned models.

# J  Visualization of Activation Magnitude of LLMs

See Figure 4 for more visualization of the magnitude of LLM layer output activations. All layer-pruned model exhibit magnitude mismatch.

Table 16: Comparison on QA benchmark with training-free methods. ‡ denotes the DeepSeek-R1-Distill version.

| Model | $L_p/L_t$ | Method | Ratio | ARC-c | ARC-e | BoolQ | HeSw | PIQA | WG | WSC | Race-h | CoPa | Avg. | RP |
|---|---|---|---|---|---|---|---|---|---|---|---|---|---|---|
| Baichuan-2-7B | 0/32 | Dense | - | 42.49 | 72.98 | 73.91 | 72.19 | 77.20 | 68.43 | 79.85 | 38.28 | 85.00 | 67.81 | 100 |
| | 9/32 | LLMPruner | 23.29 | 32.42 | 56.36 | 59.82 | 54.11 | 69.70 | 53.20 | 59.34 | 28.04 | 78.00 | 54.55 | 79.64 |
| | 9/32 | SLEB | 24.26 | 29.18 | 48.91 | 62.29 | 52.14 | 68.88 | 55.09 | 66.30 | 30.43 | 75.00 | 54.25 | 79.19 |
| | 9/32 | ShortGPT | 24.26 | 28.67 | 42.55 | 67.19 | 47.09 | 62.68 | 62.19 | 69.23 | 29.38 | 65.00 | 52.66 | 77.10 |
| | 9/32 | LLM-Streamline (None) | 24.26 | 28.67 | 42.55 | 67.19 | 47.09 | 62.68 | 62.19 | 69.23 | 29.38 | 65.00 | 52.66 | 77.10 |
| | 9/32 | LINEARPATCH[S/L] | 24.04 | 30.80 | 50.04 | 62.45 | 52.31 | 65.72 | 65.11 | 71.43 | 31.58 | 72.00 | 55.72 | **81.66** |
| | 7/32 | LLMPruner | 18.47 | 36.86 | 62.63 | 62.23 | 61.25 | 72.03 | 54.06 | 63.74 | 29.00 | 80.00 | 57.98 | 84.85 |
| | 7/32 | SLEB | 18.87 | 31.31 | 55.39 | 65.47 | 56.93 | 71.65 | 59.12 | 72.89 | 33.21 | 73.00 | 57.66 | 84.46 |
| | 7/32 | ShortGPT | 18.87 | 34.90 | 51.81 | 62.39 | 55.27 | 65.56 | 64.72 | 74.73 | 31.77 | 72.00 | 57.02 | 83.88 |
| | 7/32 | LLM-Streamline (None) | 18.87 | 34.90 | 51.81 | 62.39 | 55.27 | 65.56 | 64.72 | 74.73 | 31.77 | 72.00 | 57.02 | 83.88 |
| | 7/32 | LINEARPATCH[S/L] | 18.65 | 35.15 | 57.20 | 62.91 | 59.02 | 68.55 | 66.61 | 76.19 | 34.45 | 73.00 | 59.23 | **87.27** |
| Qwen-7B‡ | 0/28 | Dense | - | 44.62 | 66.84 | 78.38 | 60.77 | 72.42 | 60.77 | 67.03 | 36.36 | 74.00 | 62.35 | 100 |
| | 7/28 | SLEB | 21.42 | 32.17 | 52.82 | 59.42 | 46.08 | 66.97 | 53.20 | 56.41 | 23.19 | 62.00 | 50.25 | 79.39 |
| | 7/28 | ShortGPT | 21.42 | 33.62 | 58.00 | 56.24 | 45.89 | 66.97 | 53.67 | 58.61 | 30.72 | 60.00 | 51.52 | 82.58 |
| | 7/28 | LLM-Streamline (None) | 21.42 | 32.00 | 53.20 | 48.41 | 46.21 | 65.94 | 51.22 | 54.95 | 29.95 | 65.00 | 49.65 | 79.63 |
| | 7/28 | LINEARPATCH[L] | 21.25 | 32.00 | 54.84 | 60.37 | 46.57 | 68.34 | 51.85 | 55.68 | 32.15 | 69.00 | 52.31 | **83.54** |
| LLaMA-8B‡ | 0/32 | Dense | - | 42.49 | 65.91 | 82.91 | 74.35 | 77.58 | 67.80 | 82.78 | 41.53 | 89.00 | 69.37 | 100 |
| | 7/32 | SLEB | 19.01 | 34.30 | 54.38 | 62.75 | 57.27 | 70.08 | 55.33 | 66.67 | 32.92 | 76.00 | 56.63 | 81.45 |
| | 7/32 | ShortGPT | 19.01 | 37.12 | 49.03 | 77.09 | 55.95 | 66.49 | 63.14 | 70.33 | 33.21 | 71.00 | 58.15 | 83.72 |
| | 7/32 | LLM-Streamline (None) | 19.01 | 37.12 | 49.03 | 77.09 | 55.95 | 66.49 | 63.14 | 70.33 | 33.21 | 71.00 | 58.15 | 83.72 |
| | 7/32 | LINEARPATCH[L] | 18.80 | 36.52 | 53.79 | 79.72 | 60.80 | 68.23 | 66.14 | 71.06 | 37.51 | 73.00 | 60.75 | **87.69** |

Table 17: Comparison on MMLU benchmark with training-free methods.

| Model | $L_p/L_t$ | Method | STEM | Humanities | Social sciences | Others | Weighed avg. |
|---|---|---|---|---|---|---|---|
| LLaMA-2-13B | 0/40 | Dense | 44.14 | 54.35 | 63.44 | 60.80 | 55.63 |
| | 8/40 | ShortGPT | 42.80 | 50.13 | 62.78 | 61.19 | 53.88 |
| | 8/40 | +LINEARPATCH | 43.07 | 50.61 | 62.56 | 61.04 | **54.01** |
| | 8/40 | LLM-Streamline (None) | 42.80 | 50.13 | 62.78 | 61.19 | 53.88 |
| | 8/40 | +LINEARPATCH | 43.07 | 50.61 | 62.56 | 61.04 | **54.01** |
| LLaMA-3-8B | 0/32 | Dense | 55.20 | 59.00 | 75.95 | 71.56 | 64.80 |
| | 5/32 | ShortGPT | 46.92 | 53.92 | 65.65 | 65.42 | 57.64 |
| | 5/32 | +LINEARPATCH | 44.67 | 50.31 | 65.91 | 61.91 | **55.19** |
| | 5/32 | LLM-Streamline (None) | 53.47 | 56.08 | 74.58 | 68.32 | 62.40 |
| | 5/32 | +LINEARPATCH | 54.24 | 57.15 | 75.40 | 71.50 | **63.84** |
| Baichuan2-7B | 0/32 | Dense | 44.53 | 51.30 | 61.23 | 60.85 | 54.23 |
| | 7/32 | ShortGPT | 42.01 | 45.48 | 58.17 | 55.71 | 49.88 |
| | 7/32 | +LINEARPATCH | 42.84 | 48.42 | 59.60 | 58.70 | **52.00** |
| | 7/32 | LLM-Streamline (None) | 42.01 | 45.48 | 58.17 | 55.71 | 49.88 |
| | 7/32 | +LINEARPATCH | 42.84 | 48.42 | 59.60 | 58.70 | **52.00** |
| LLaMA-2-7B | 0/32 | Dense | 36.98 | 43.25 | 51.77 | 52.47 | 45.90 |
| | 7/32 | ShortGPT | 31.75 | 37.90 | 44.72 | 46.18 | 39.98 |
| | 7/32 | +LINEARPATCH | 31.71 | 39.26 | 45.82 | 47.07 | **40.88** |
| | 7/32 | LLM-Streamline (None) | 31.75 | 37.90 | 44.72 | 46.18 | 39.98 |
| | 7/32 | +LINEARPATCH | 31.71 | 39.26 | 45.82 | 47.07 | **40.88** |

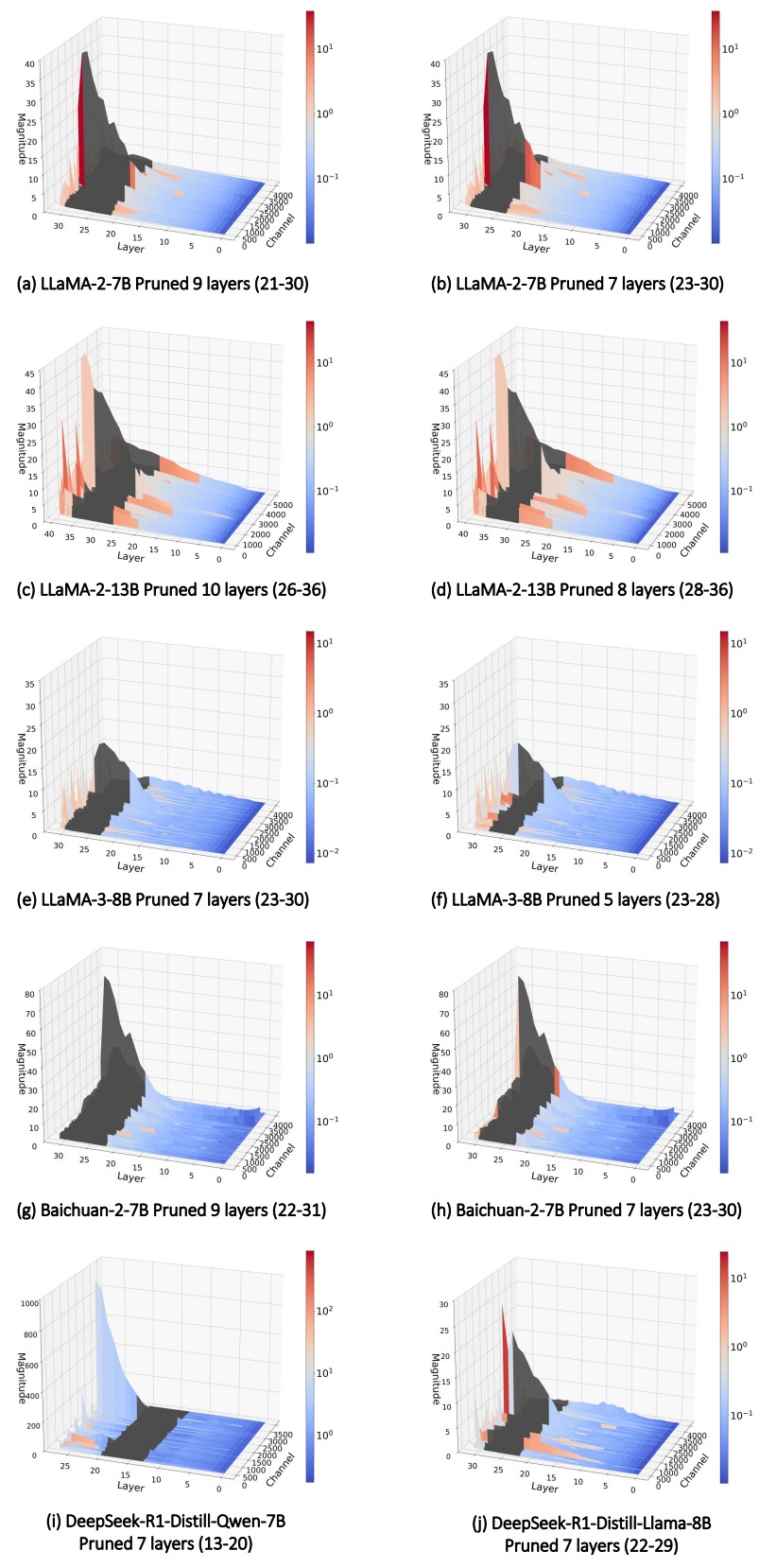

Figure 4: Visualization of the magnitude of LLM layer output activations, where pruned layers of LINEARPATCH$_{[L]}$ are represented in grey. All layer-pruned model exhibit magnitude mismatch.

