# OpenReview forum: "A Simple Linear Patch Revives Layer-Pruned Large Language Models"
_NeurIPS.cc/2025/Conference — NeurIPS 2025 poster_

### Official Review · Reviewer_e7ph · 2025-06-28

**Clarity:** 2
**Significance:** 3
**Originality:** 3
**Rating:** 5
**Confidence:** 5

**Summary:**

This work identifies activation magnitude mismatch before/after pruning site as a cause for performance drop observed during layer pruning. The authors proposed Hadamard Transofmration as a remedy. The proposed technique was extensively validated and well ablated.

**Questions:**

Can you comment on the mechanism with which Hadamard transformation can mitigate outlier activations? I think this is assumed knowledge in this paper, but is worth explaining to make this work self-contained.

Reading the ablation studies, it seems like this Hadamard transformation is not really introducing significant benefits (mostly helps language modling metrics, not downstream task performance). It would be interesting to see this technique without Hadamard transformation, but finetuned. I believe the significance of this work will be further enhanced if the solution/mitigation can be further simplified.

**Ethical Concerns:**

["NO or VERY MINOR ethics concerns only"]

**Final Justification:**

I appreciate the authors' thorough follow-up and my questions and concerns are addressed. I maintain my recommendation of acceptance.

**Limitations:**

yes

**Quality:**

3

**Strengths And Weaknesses:**

Strength:
- Compelling motivation and analysis of the pathology from layer pruning.
- The designed method is simple and effective.
- The authors presented compelling experimental evaluation, as well as thorough ablation.
- The subject of study -- layer pruning -- is conceptually simple and easy to deploy and has potential for wide real world impact.
- Overall I think this paper presents a solid engineering contribution.

Weakness:
- Biggest weakness is writing -- for example, the phrase "pruning interface" is used without definition; in abstract, the author claims that they can remove 5 layers from llama-3 8B model, but does not mention how many layers in total are there in this model.
- Appears that the pruning regime studied is fairly conservative (<=30%), which limits the significance of the proposed technique in real-world settings.

---

> ### Author Rebuttal · Authors · 2025-07-27
>
> Thanks for your invaluable reviews. We provide point-by-point responses below.
>
> >Q1: Biggest weakness is writing -- for example, the phrase "pruning interface" is used without definition; in abstract, the author claims that they can remove 5 layers from llama-3 8B model, but does not mention how many layers in total are there in this model.
>
> A1: We sincerely appreciate the reviewer's feedback on the manuscript's clarity.
> * To address the concerns, we have revised the terminology and provided explicit definitions for technical terms such as "pruning interface" in the context.
> * Additionally, we have included the total number of layers in the Llama-3 8B model (32 layers) when discussing the removal of 5 layers, both in the abstract and relevant sections, improving the reader's understanding of the experimental setup.
>
> >Q2: Appears that the pruning regime studied is fairly conservative (<=30%), which limits the significance of the proposed technique in real-world settings.
>
> A2: Unlike other pruning methods, **layer pruning exhibits an intuitive positive correlation between pruning ratio and speedup**, and does not rely on additional software or hardware support (such as CUDA operators or device computing cores), offering advantages in practical applications.
>
> * Structured sparsity like 2:4 sparsity, even with a **50% pruning ratio**, achieves only a **1.24× speedup** on LLaMA-7B and requires specific hardware devices [1].
>
> * Layer pruning with a **30% pruning ratio** can reduce latency by nearly 30%, equivalent to a **1.43× speedup**.
>
> Furthermore, this compression ratio strikes a balance between practical acceleration and model performance. For instance, when the pruning ratio of LLaMA-2-13B is 24.17%, it retains 90.49% of the QA performance.
>
> ---
> [1] Sun, M., Liu, Z., Bair, A., & Kolter, J. Z. A Simple and Effective Pruning Approach for Large Language Models. In The Twelfth International Conference on Learning Representations.
>
>
> >Q3: Can you comment on the mechanism with which Hadamard transformation can mitigate outlier activations? I think this is assumed knowledge in this paper, but is worth explaining to make this work self-contained.
>
> A3: We thank the reviewer for highlighting the mechanism of Hadamard transformation.
>
> * In the revised manuscript, we have added detailed explanation of mechanism of Hadamard transformation in the Hadamard Transformation paragraph of Section 3.3. Briefly, multiplication with the Hadamard matrix encourages a more balanced distribution of activation across channels.
>
> * We provide mathematical proofs below.
> A symmetric Hessian matrix $H\in\mathbb{R}^{n\times n}$ is μ-incoherent if it has an eigendecomposition $H=Q\Lambda Q^{\top}$ such that for all $i$ and $j$, $|Q_{ij}|=|e_{i}^{\top}Qe_{j}|\leq\mu/\sqrt{n}$. One straightforward way to make a symmetric matrix incoherent is to conjugate it by a uniform random orthogonal matrix: this will result in each of its eigenvectors being a random unit vector, whose entries will concentrate around magnitude $n^{-1/2}$. [2][3]
>
> In this work, we introduce incoherence processing into activation with Hadamard transformation. A activation matrix $X\in\mathbb{R}^{n\times m}$is μ-incoherent if all $i$ and $j$,  $|X_{ij}|=|e_{i}^{T}We_{j}|  \leq\mu/||X||_F \sqrt{nm}.$.
>
> Then, we obtain:
>
> $max(X) \leq\mu||X||_F /\sqrt{nm},$
>
> where $max$ is the element-wise max of the matrix, and $mn$ is the number of elements.
>
> Incoherence in activation $X$ can be viewed as a form of outlier reduction: a small bound on the magnitude of its entries across tokens means that we do not need to tailor a specific scaling parameter for each token and therefore eliminate the token-wise magnitude mismatch.
>
> ---
>
> [2] Ashkboos, S., Mohtashami, A., Croci, M. L., Li, B., Cameron, P., Jaggi, M., ... & Hensman, J. (2024). Quarot: Outlier-free 4-bit inference in rotated llms. Advances in Neural Information Processing Systems, 37, 100213-100240.
>
> [3] Chee, J., Cai, Y., Kuleshov, V., & De Sa, C. M. (2023). Quip: 2-bit quantization of large language models with guarantees. Advances in Neural Information Processing Systems, 36, 4396-4429.
>
>
>
>
> > Q4: Reading the ablation studies, it seems like this Hadamard transformation is not really introducing significant benefits (mostly helps language modling metrics, not downstream task performance). It would be interesting to see this technique without Hadamard transformation, but finetuned. I believe the significance of this work will be further enhanced if the solution/mitigation can be further simplified.
>
> A4:
> * Ablation Study on Hadamard Transformation
>
> We have supplemented the ablation experimental results based on the LLaMA-3-8B model in Table 1, which confirm that the Hadamard transformation remains effective in downstream task like QA benchmark. Under the training-free configuration, the Hadamard transformation brings a 1.05% relative performance improvement, and this gain is of practical value. It should be noted that the equivalent transformations formed by Hadamard orthogonal matrices do not affect performance, while the core goal of introducing the Hadamard transformation is to achieve the smoothing of activations.
>
> ---
> Table 1: Ablation study on the ingredients of LINEARPATCH over LLaMA-3-8B with 5 out of 32 layers pruned. +d applies channel scaling, +P(i.e., ${HDH^\top}$) refers to the LINEARPATCH.
> |  | ARC-c | ARC-e | BoolQ | HeSw | PIQA | WG | WSC | Race-h | CoPa | AVG | RP |
> |---|---|---|---|---|---|---|---|---|---|---|---|
> | Dense | 53.41 | 77.78 | 81.28 | 79.16 | 80.85 | 72.85 | 86.45 | 40.19 | 89.00 | 73.44 | 100.00 |
> | Vanilla | 47.35 | 66.20 | 73.52 | 71.10 | 74.27 | 71.03 | 76.56 | 36.65 | 84.00 | 66.74 | 90.84 |
> |  +d | 48.63 | 69.99 | 74.07 | 72.63 | 75.95 | 72.61 | 79.12 | 37.51 | 85.00 | 68.39 | 93.10 (+2.26%) |
> |  +P | 48.55 | 70.71 | 74.25 | 72.52 | 76.71 | 73.95 | 81.32 | 38.37 | 86.00 | 69.15 | 94.15 (+1.05%) |
>
>
> * Explanation of Finetuned Parameter
>
> Further simplifying the approach is worthy of exploration and will also be the focus of our subsequent research. In the current study, we found that when only the scaling parameters are trained while removing the Hadamard transformation, the model performance will be at a suboptimal level (see Table 2 for details). In future research, we plan to integrate the Hadamard transformation into the model weights through equivalent transformations, thereby reducing the computational complexity of this part.
>
> ---
> Table 2: Ablation on finetuned  parameters over LLaMA-2-7B with 9 out of 32 layers pruned. RP denotes the retained performance. d denotes channel scaling, +P(i.e., ${HDH^\top}$) refers to the LINEARPATCH.
>
> | Parameters | WIKI-2 | C4 | PTB | PPL avg. | QA RP |
> |---|---|---|---|---|---|
> | w/o (Dense) | 5.47 | 6.97 | 22.51 | 11.65 | 100 |
> | w/o (Vanilla) | 35.68 | 36.10 | 96.52 | 56.10 | 80.29 |
> |  d | 17.51  | 18.64  | 51.64  | 29.26 | 84.45 (+4.16%)|
> |  P | 8.60  | 12.98  | 37.16  | 19.58 | 88.15 (+7.86%)|

---

> > ### Comment · Reviewer_e7ph · 2025-08-04
> > **Ack**
> >
> > I appreciate author's detailed rebuttal to address my concerns. I have read the author response and decide to maintain my recommendation for accpetance.

---

### Official Review · Reviewer_9aSJ · 2025-06-28

**Clarity:** 3
**Significance:** 2
**Originality:** 3
**Rating:** 2
**Confidence:** 3

**Summary:**

This paper proposes LINEARPATCH, a simple and plug-and-play method for pruning large language models (LLMs). The paper highlights that the mismatch in channel magnitudes directly impacts the performance of pruned LLMs and proposes a dimension-wise scaling $d$. Considering that massive outliers specific to particular tokens further degrade pruning effectiveness, this paper uses Hadamard transformations $H$. Finally, a Memory-Efficient Offline Knowledge Distillation strategy is employed to further enhance the performance. Experimental results demonstrate that, compared to baselines, LINEARPATCH achieves superior performance in preserving accuracy and computational efficiency.

**Questions:**

See weaknesses.

**Ethical Concerns:**

["NO or VERY MINOR ethics concerns only"]

**Final Justification:**

I acknowledge the methodological contributions of this paper; my concerns mainly focus on the evaluation. This paper lacks many of the datasets used in baselines, and the experimental results are inconsistent, which raises concerns about the effectiveness of the methods presented in this paper.

**Limitations:**

yes

**Quality:**

2

**Strengths And Weaknesses:**

Strengths:
1. The paper is well-written.
2. The proposed method is applicable to both training-free settings and fine-tuning.
3. In simple terms, the paper highlights the problem of norm inconsistency after pruning and introduces a method for initializing the parameter matrix. Dimension-wise scaling and the proposed method are both intuitive.

Weaknesses:
1. As shown in Tables 15 and 16, the proposed method often fails to outperform the baselines. Additionally, the datasets and results in Table 1 differ from those reported in ShortGPT and LLM-Streamline. Therefore, I have serious doubts about the effectiveness of the proposed method.
2. In the fine-tuning setting, does the proposed method outperform a randomly initialized P?
3. In Table 1, why are the results of ShortGPT and LLM-Streamline identical for LLaMA-2-7B and LLaMA-2-13B, but different for LLaMA-3-8B?


Minor:
1. The left side of Figure 1(a) appears somewhat cluttered. Given the significant magnitude differences, the alignment lines seem unnecessary and make the overall figure cluttered.

2. Changing "channel" to "dimension" may also improve readability.

---

> ### Author Rebuttal · Authors · 2025-07-28
>
> Thanks for your reviews. We provide point-by-point responses below.
>
> >Q1: As shown in Tables 15 and 16, the proposed method often fails to outperform the baselines. Additionally, the datasets and results in Table 1 differ from those reported in ShortGPT and LLM-Streamline. Therefore, I have serious doubts about the effectiveness of the proposed method.
>
> A1:
>
> **Baseline Comparison Issues**
>
> 1. **Plug-and-play method**. We highlight that LinearPatch is a plug-and-play method that can be combined with different pruning metrics for further improvement. By **reorganzing the results** of the corresponding baselines from Table 11, 12, 13, and 15, it can be found the **consistent improvement** of LinearPatch relative to baselines, as shown below in Table A and Table B.
>
> 2. **Corresponding baselines**. The baseline of ${LinearPatch_{[L]}}$ is LLM-Streamline (None), and the baseline of ${LinearPatch_{[S]}}$ is ShortGPT.   We have claimed this at the beginning of Section 4.3 Main Results (lines 216-218). Therefore, the proposed method **always outperform the baselines in Tables 15 and mostly outperform the baselines in Table 16 as presented in Table A, Table B and Table C.**
>
> 3. **The overfitting of SLEB**. Tables 15 and 16 are placed together to illustrate that SLEB [1] with PPL pruning metric tends to overfit, resulting in suboptimal performance on QA and MMLU tasks, as also claimed in Appendix G.2 (line 494).
>
>
> **Reproducibility Issues**
>
> The original papers of ShortGPT, LLM-Streamline and  Shortened LLaMA [2-4] do not claim the versions of lm_eval for the evaluation, as different versions of lm_eval may influence the results. For fair comparisons, **we consistently adopt lm_eval (v0.4.4) throughout our experiments**, and used **the same models and pruning indices as the baseline methods to ensure the reproduction.** All the configurations and code will be released for the community.
>
> ---
>
> [1] Song, J., Oh, K., Kim, T., Kim, H., Kim, Y., & Kim, J. J. (2024, July). SLEB: Streamlining LLMs through Redundancy Verification and Elimination of Transformer Blocks. In International Conference on Machine Learning (pp. 46136-46155). PMLR.
>
> [2] Men, X., Xu, M., Zhang, Q., Wang, B., Lin, H., Lu, Y., ... & Chen, W. (2024). Shortgpt: Layers in large language models are more redundant than you expect. arXiv preprint arXiv:2403.03853.
>
> [3] Chen, X., Hu, Y., Zhang, J., Wang, Y., Li, C., & Chen, H. (2024). Streamlining redundant layers to compress large language models. arXiv preprint arXiv:2403.19135.
>
> [4] Kim, B. K., Kim, G., Kim, T. H., Castells, T., Choi, S., Shin, J., & Song, H. K. (2024). Shortened llama: A simple depth pruning for large language models. arXiv preprint arXiv:2402.02834, 11, 1.
>
>
> ---
>
>
> **Tabel A.  Comparison on PPL benchmark over LLaMA-2-7B with 9 out of 32 layers pruned. (reorganized)**
>
> | Method | WIKI-2 | C4 | PTB | AVG |
> |---|---|---|---|---|
> | Dense | 5.47  | 6.97  | 22.51  | 11.65  |
> | SLEB | 9.14  | 11.21  | 38.45  | 19.60  |
> | +$LinearPatch$ | 8.77  | 10.66  | 38.30  | 19.24  |
> | Taylor+ | 18.45  | 20.99  | 62.18  | 33.87  |
> | +$LinearPatch$ | 13.84  | 15.28  | 48.26  | 25.79  |
> | ShotGPT | 18.45  | 20.99  | 62.18  | 33.87  |
> | +$LinearPatch$ | 13.22  | 14.58  | 45.97  | 24.59  |
> | LLM-Streanline(None) | 18.45  | 20.99  | 62.18  | 33.87  |
> | +$LinearPatch$ | 13.22  | 14.58  | 45.97  | 24.59  |
>
> ---
>
> **Tabel B.  Comparison on PPL benchmark over LLaMA-3-8B with 7 out of 32 layers. (reorganized)**
>
>
> | Method | WIKI-2 | C4 | PTB | AVG |
> |---|---|---|---|---|
> | Dense | 6.14  | 8.88  | 10.59  | 8.54  |
> | SLEB | 13.12  | 16.76  | 21.04  | 16.97  |
> | +$LinearPatch$ | 11.97  | 15.74  | 19.55  | 15.75  |
> | Taylor+ | 2287.86  | 1491.38  | 4741.90  | 2840.38  |
> | +$LinearPatch$ | 208.88  | 235.63  | 264.97  | 236.49  |
> | ShotGPT | 57.76  | 50.13  | 67.39  | 58.43  |
> | +$LinearPatch$ | 25.67  | 28.38  | 31.22  | 28.42  |
> | LLM-Streanline(None) | 2287.73  | 1491.37  | 4738.81  | 2839.30  |
> | +$LinearPatch$ | 69.82  | 96.68  | 88.79  | 85.10  |
>
>
> ---
>
>
> **Tabel C.  Comparison on MMLU benchmark. (reorganized)**
>
> | Model | Puned/total layers | Method | STEM | Humanities | Social sciences | Others | Weighed average accuracy |
> |---|---|---|---|---|---|---|---|
> | LLaMA-2-13b | 0/40 | Dense | 44.14% | 54.35% | 63.44% | 60.80% | 55.63% |
> |  | 8/40 | ShotGPT | 42.80% | 50.13% | 62.78% | 61.19% | 53.88% |
> |  | 8/40 | +$LinearPatch$ | 43.07% | 50.61% | 62.56% | 61.04% | 54.01% |
> |  | 8/40 | LLM-Streanline(None) | 42.80% | 50.13% | 62.78% | 61.19% | 53.88% |
> |  | 8/40 | +$LinearPatch$ | 43.07% | 50.61% | 62.56% | 61.04% | 54.01% |
> | LLaMA-3-8b | 0/32 | Dense | 55.20% | 59.00% | 75.95% | 71.56% | 64.80% |
> |  | 5/32 | ShotGPT  | 46.92% | 53.92% | 65.65% | 65.42% | 57.64% |
> |  | 5/32 | +$LinearPatch$ | 44.67% | 50.31% | 65.91% | 61.91% | 55.19% |
> |  | 5/32 | LLM-Streanline(None) | 53.47% | 56.08% | 74.58% | 68.32% | 62.40% |
> |  | 5/32 | +$LinearPatch$ | 54.24% | 57.15% | 75.40% | 71.50% | 63.84% |
> | Baichuan2-7b | 0/32 | Dense | 44.53% | 51.30% | 61.23% | 60.85% | 54.23% |
> |  | 7/32 | ShotGPT | 42.01% | 45.48% | 58.17% | 55.71% | 49.88% |
> |  | 7/32 | +$LinearPatch$ | 42.84% | 48.42% | 59.60% | 58.70% | 52.00% |
> |  | 7/32 | LLM-Streanline(None) | 42.01% | 45.48% | 58.17% | 55.71% | 49.88% |
> |  | 7/32 | +$LinearPatch$ | 42.84% | 48.42% | 59.60% | 58.70% | 52.00% |
> | LLaMA-2-7b | 0/32 | Dense | 36.98% | 43.25% | 51.77% | 52.47% | 45.90% |
> |  | 7/32 | ShotGPT | 31.75% | 37.90% | 44.72% | 46.18% | 39.98% |
> |  | 7/32 | +$LinearPatch$ | 31.71% | 39.26% | 45.82% | 47.07% | 40.88% |
> |  | 7/32 | LLM-Streanline(None) | 31.75% | 37.90% | 44.72% | 46.18% | 39.98% |
> |  | 7/32 | +$LinearPatch$ | 31.71% | 39.26% | 45.82% | 47.07% | 40.88% |
>
>
> ---
>
>
> >Q2: In the fine-tuning setting, does the proposed method outperform a randomly initialized P?
>
> A2: **LinearPatch addresses the issue of layer-wise and token-wise mismatch in activation magnitudes**, and can quickly restore performance combined with lightweight training. **In contrast, inserting a random matrix into the LLM exacerbates such mismatch.** A randomly initialized P makes layer-pruned LLM diverge with loss NAN during the output distillation fine-tuning process.  In comparison, our work offers an effective initialization method for the patch.
>
>
>
>
> >Q3: In Table 1, why are the results of ShortGPT and LLM-Streamline identical for LLaMA-2-7B and LLaMA-2-13B, but different for LLaMA-3-8B?
>
> A3:
>
> ShortGPT [2] and LLM-Streamline [3]  share the same set of pruned layers, and thus have the same performance. For LLaMA-2-7B and LLaMA-2-13B, the pruning indices of the two methods are the same, whereas they differ for LLaMA-3-8B. We have reported the pruned layer indices of these methods in Appendix A.
>
>
> >Q4: The left side of Figure 1(a) appears somewhat cluttered. Given the significant magnitude differences, the alignment lines seem unnecessary and make the overall figure cluttered.
>
> A4: We thank the reviewer for pointing out the clarity issue in Figure 1(a). The alignment lines were intended to illustrate the gap before and after alignment, as well as the effect of alignment. We believe these auxiliary lines effectively help readers understand the process of activation scaling.
>
>
> >Q5: Changing "channel" to "dimension" may also improve readability.
>
> A5: In the manuscript, **the term "channel" follows the conventions established in references [5]–[10], which are widely cited in the model compression community.** This term is commonly used in structured pruning techniques, such as "channel-wise pruning" described in [5], and "scale activations  described channel-wise" in [8].
>
> ---
>
>
> [5] Ma, X., Fang, G., & Wang, X. (2023). Llm-pruner: On the structural pruning of large language models. Advances in neural information processing systems, 36, 21702-21720.
>
> [6] Nrusimha, A., Mishra, M., Wang, N., Alistarh, D., Panda, R., & Kim, Y. (2024). Mitigating the impact of outlier channels for language model quantization with activation regularization. arXiv preprint arXiv:2404.03605.
>
> [7] Sun, M., Liu, Z., Bair, A., & Kolter, J. Z. (2023). A Simple and Effective Pruning Approach for Large Language Models. In The Twelfth International Conference on Learning Representations.
>
> [8] Frantar, E., & Alistarh, D. (2023, July). Sparsegpt: Massive language models can be accurately pruned in one-shot. In International conference on machine learning (pp. 10323-10337). PMLR.
>
> [9] Zhang, Y., Bai, H., Lin, H., Zhao, J., Hou, L., & Cannistraci, C. V. (2024). Plug-and-Play: An Efficient Post-training Pruning Method for Large Language Models. In The Twelfth International Conference on Learning Representations.
>
> [10] Ashkboos, S., Mohtashami, A., Croci, M. L., Li, B., Cameron, P., Jaggi, M., ... & Hensman, J. (2024). Quarot: Outlier-free 4-bit inference in rotated llms. Advances in Neural Information Processing Systems, 37, 100213-100240.

---

> > ### Comment · Reviewer_9aSJ · 2025-08-04
> >
> > Thank you for your reply.
> > 1. I believe my concerns have not been fully addressed. The main results in the paper differ significantly from previous baselines in terms of both datasets used and reported performance, making it difficult for me to assess the validity of the proposed method, even though it appears intuitive.
> > 2. Moreover, since P can be fine-tuned, whether the proposed initialization strategy for P can consistently outperform random initialization after fine-tuning remains an open and worthwhile question for further investigation.
> > 3. Lastly, I suggest the authors resolve the reproducibility issues of the baselines via email or GitHub. Large performance gaps and known reproducibility problems in prior methods weaken the persuasiveness of the paper.

---

> > > ### Author Response · Authors · 2025-08-07
> > > **Reminder**
> > >
> > > Dear Reviewer 9aSJ,
> > >
> > > I hope this message finds you well. As the discussion period is drawing to a close with less than two days remaining, I wanted to ensure we have addressed all your concerns satisfactorily, especially your concern about reproducibility of baselines. If there are any additional points or feedback you'd like us to consider, please let us know. Your insights are invaluable to us, and we're eager to address any remaining issues to improve our work.
> > >
> > > Thank you for your time and effort in reviewing our paper.

---

> > > ### Comment · Area_Chair_pfQN · 2025-08-08
> > >
> > > Dear Reviewers,
> > >
> > > This is a gentle reminder to please continue participating in the discussion phase, especially addressing the points raised by the authors. Thanks!
> > >
> > > Best,
> > >
> > > AC

---

> ### Author Response · Authors · 2025-08-06
> **Response to Reviewer 9aSJ**
>
> >Q1: The main results in the paper differ significantly from previous baselines in terms of both datasets used and reported performance, making it difficult for me to assess the validity of the proposed method, even though it appears intuitive.
>
>
> A1: **We highlight that it is an objective challenge to extactly reproduce existing baselines, and this objective challenge should not be considered as the factor to downgrade our work**. In fact, **even for existing published works [1,2,3,4,5] , the results often differ from one another, as shown in Table A** (extracted from their original papers). As we mentioned in our earlier reply, **the key challenge behind is that the version of lm_eval is not explictly mentioned in piror works.**
>
> Nonetheless, we have made every effort to make the results reproducible. We fairly compare all differnet baselins with the same version of lm_eval (v 0.4.4). We will also release our code and evaluation script for easy reproducibility for the research community.
>
> ---
>
> Table A: Reproduction results of SliceGPT.
>
>  |  | HellaSwag | PIQA | WinoGrande | ARC-e | ARC-c | CoQA | Location |
> |---|---|---|---|---|---|---|---|
> | SliceGPT | 54.16 | 66.87 | 63.38 | 58.46 | 34.56 | - | Table 7, [1] |
> | Reproduction by ShotGPT | 58.1 | 68.55 | 62.04 | 56.15 | 35.07 | 49.6 | Table 8, [2] |
> | Reproduction by LLM-Streamline | 47.5 | 68.3 | - | - | - | 41.36 | Table 2, [3] |
> | Reproduction by LaCO | 50.27 | 66.21 | - | - | - | 41.36 | Table 1, [4] |
> | Reproduction by SLEB | 54.16 | 66.76 | 63.38 | 58.42 | 34.64 | - | Table 9, [5] |
>
> ---
>
> [1] Ashkboos, S., Croci, M. L., Nascimento, M. G. D., Hoefler, T., & Hensman, J. (2024). Slicegpt: Compress large language models by deleting rows and columns. arXiv preprint arXiv:2401.15024.
>
> [2] Men, X., Xu, M., Zhang, Q., Wang, B., Lin, H., Lu, Y., ... & Chen, W. (2024). Shortgpt: Layers in large language models are more redundant than you expect. arXiv preprint arXiv:2403.03853.
>
> [3] Chen, X., Hu, Y., Zhang, J., Wang, Y., Li, C., & Chen, H. (2024). Streamlining redundant layers to compress large language models. arXiv preprint arXiv:2403.19135.
>
> [4] Yang, Y., Cao, Z., & Zhao, H. (2024). Laco: Large language model pruning via layer collapse. arXiv preprint arXiv:2402.11187.
>
> [5] Song, J., Oh, K., Kim, T., Kim, H., Kim, Y., & Kim, J. J. (2024, July). SLEB: Streamlining LLMs through Redundancy Verification and Elimination of Transformer Blocks. In International Conference on Machine Learning (pp. 46136-46155). PMLR.
>
> ---
>
> >Q2: Moreover, since P can be fine-tuned, whether the proposed initialization strategy for P can consistently outperform random initialization after fine-tuning remains an open and worthwhile question for further investigation.
>
> A2: As shown in Table B, **fine-tuning the randomly initialized P cases the model to collapse**. The reason is that, as explained in our earlier reply, inserting a random matrix into the LLM exacerbates the layer-wise and token-wise mismatch in activation magnitudes, while the proposed LinearPatch can effectively address this issue.
>
> ___
>
>
> Table B: Comparison of LinearPatch initialization in finetuning case on Perplexity benchmark on LLaMA-2-7B with 9 out of 32 layers pruned.
>
> |  | WIKI-2 | C4 | PTB | PPL avg. |
> |---|---|---|---|---|
> | w/o (Vanilla) | 35.68 | 36.10 | 96.52 | 56.10 |
> |  Random Initialized | 3798.19 | 20318.69 | 11427.18 | 11848.02 |
> |  P | **8.60**  | **12.98**  | **37.16**  | **19.58** |
>
> ---
>
> >Q3: Lastly, I suggest the authors resolve the reproducibility issues of the baselines via email or GitHub. Large performance gaps and known reproducibility problems in prior methods weaken the persuasiveness of the paper.
>
> A3: **Providing links is not permitted during the discussion.** Nonetheless, we will open source our code and evaluation script for reproducibility of our research, as stated in A1.

---

> > ### Comment · Reviewer_9aSJ · 2025-08-08
> >
> > Thanks for your reply. I acknowledge the methodological contributions of this paper; my concerns mainly focus on the evaluation. Specifically,
> > 1. In my previous response, I already mentioned the dataset issue. For example, shortGPT uses the CMNLI, HeSw, PIQA, CHID, CoQA, BoolQ, Race-H, Race-M, C3, MMLU, and CMMLU datasets, while LLM-Streamline further includes the WSC dataset. The datasets used in this paper appear incomplete, with several missing. Given the inconsistencies in the results, I am therefore concerned about the effectiveness of this method, even though the authors have clarified some points.
> >
> > 2. In Lines 441–442, the MMLU dataset does not appear to have a version issue with $lm_{eval}$. However, the proposed method fails to effectively outperform the baseline methods.
> >
> > Finally, I raised some of the scores.

---

> ### Author Response · Authors · 2025-08-09
> **Response to Reviewer 9aSJ**
>
> We sincerely thank reviewer 9aSJ for the continued engagement and for raising the score. Nevertheless, we would like to offer the last clarification before the discussion closes.
>
> > Q1. Dataset issues
>
> The QA benchmarks are diverse, yet details on how prior methods evaluate datasets such as CMNLI, CHID, and C3 remain unclear. Moreover, these datasets fall outside the scope of $lm_{eval}$ library, which may hinder reproducibility and fair comparisons. We nevertheless appreciate the reviewer’s suggestion and will consider including additional benchmarks in future work.
>
> > Q2. The proposed method underperforms baselines on MMLU
>
> A2: **It is improper to assert that our method underperforms baselines on MMLU**. According to Table 16, LinearPatch surpasses baselines in 10 out of 16 cases (formed by four models, two sparsity ratios, and two pruning metrics), a result that is statistically favorable.
>
> **We believe a fair assessment should rely on the full empirical picture rather than on a single dataset**. On most benchmarks—including WIKI-2, C4, PTB, ARC-c, ARC-e, HeSw, PIQA, WSC, Race-h, and CoPa—our method consistently outperforms baselines (16/16 cases). The margin narrows on BoolQ (10/16), WG (9/16), and MMLU (10/16), yet the overall trend remains positive. We note that any novel method may exhibit varying degrees of advantage across tasks; what matters is the aggregate benefit, which LinearPatch clearly delivers.
>
> Table A: LinearPatch win counts over 16 experimental configurations (LLaMA-2-7B/13B, LLaMA-3-8B, Baichuan2-7B, two sparsity ratios, two pruning metrics).
> |                          | WIKI-2 | C4    | PTB   | ARC-c | ARC-e | HeSw  | PIQA  | WSC   | Race-h | CoPa  | BoolQ | WG   | MMLU  |
> |--------------------------|--------|-------|-------|-------|-------|-------|-------|-------|--------|-------|-------|------|-------|
> | wining rate | 16/16  | 16/16 | 16/16 | 16/16 | 16/16 | 16/16 | 16/16 | 16/16 | 16/16  | 16/16 | 10/16 | 9/16 | 10/16 |

---

### Official Review · Reviewer_oSnA · 2025-06-28

**Clarity:** 3
**Significance:** 3
**Originality:** 3
**Rating:** 4
**Confidence:** 4

**Summary:**

This paper proposes LINEARPATCH, a simple and efficient plug-and-play method to restore the performance of layer-pruned large language models by addressing activation magnitude mismatches at the pruning interface, using a combination of Hadamard transformation and channel-wise scaling, and further improves results through lightweight offline knowledge distillation—demonstrating strong gains across multiple LLM architectures and benchmarks with minimal inference overhead.

**Questions:**

While the empirical benefits of Hadamard transformation and scaling are clear, can you provide more intuition or theoretical justification for why this combination is effective for aligning activation magnitudes after pruning? Additionally, are there identified scenarios where this approach fails or is suboptimal?

**Ethical Concerns:**

["NO or VERY MINOR ethics concerns only"]

**Limitations:**

yes, authors addressed the limitations and potential negative societal impact of their work

**Quality:**

3

**Strengths And Weaknesses:**

Strength:

1.LINEARPATCH is a lightweight, plug-and-play method that combines Hadamard transformation and channel-wise scaling into a single matrix, making it easy to implement and introducing negligible inference overhead.

2.LINEARPATCH is shown to be agnostic to the pruning metric and compatible with multiple pruning strategies, and can be further improved with efficient, memory-light knowledge distillation.

3.The method is extensively evaluated on multiple LLM architectures (e.g., LLaMA-2/3, Baichuan, DeepSeek) and various tasks (QA, PPL, MMLU), consistently outperforming prior state-of-the-art layer pruning approaches both in zero-shot and post-training scenarios.


Weakness:

1.The channel-wise scaling parameters rely on a calibration set. The impact of poor or domain-mismatched calibration data on final performance is not fully analyzed.

2.The paper briefly acknowledges that certain tasks, especially those requiring complex reasoning, might still be more affected by layer pruning, but does not provide a systematic analysis or mitigation strategies.

3.While the empirical results are strong, the paper does not provide theoretical analysis on why Hadamard transformation and scaling are sufficient for all types of activation mismatches.

---

> ### Author Rebuttal · Authors · 2025-07-27
>
> **Thank you for your positive review and constructive comments. We have performed supplemented experiments and will include them in the camera-ready version.**
>
> >Q1: The channel-wise scaling parameters rely on a calibration set. The impact of poor or domain-mismatched calibration data on final performance is not fully analyzed.
>
> A1: We sincerely appreciate the reviewer's insightful comment. In layer pruning, WikiText-2 is a commonly used calibration dataset. **To thoroughly investigate the impact of different calibration datasets and validate the robustness of our method, we have supplemented experimental results using C4 and PTB as calibration sets. The results are presented in Table 1.**
>
> * Datasets Used in Experiments:
>
> 1. WikiText-2: Extracted from Wikipedia's "Good" and "Featured" articles, it is one of the most widely used and reliable small-scale benchmarks in current language modelling evaluations. It has **high quality** due to manual verification and cleaning.
>
> 2. PTB (Penn Treebank): Small in scale (approximately 1 million words), narrow in domain (Wall Street Journal articles from the 1980s), outdated, and limited in vocabulary, resulting in **moderate quality.**
>
> 3. C4: Filtered from Common Crawl, it is large in scale but contains a significant amount of repetitive and low-quality web text, suffering from high noise, leading to relatively **low quality.**
>
> ---
>
> Table 1: Ablation on calibration datasets over LLaMA-2-7B with 9 out of 32 layers pruned.
> | Calibration set | Data quality | WIKI-2 | PTB | C4 | PPL avg. |
> |---|---|---|---|---|---|
> | WIKI-2 | best | **18.6** | 53.00 | 19.28 | **30.29** |
> | PTB | mediate | 19.31 | **52.24** | 19.66 | 30.40 |
> | C4 | poor | 19.11 | 53.21 | **19.25** | 30.52 |
>
> * Key Findings:
>
> 1. **Domain-Specific Calibration Boosts Performance**: When PTB is used as the calibration set, the perplexity (PPL) on PTB drops to 52.25, significantly outperforming other calibration sets. This indicates that calibrating with domain-specific data enhances performance in that domain.
>
> 2. **Robustness to Data Quality**: Despite C4 being the lowest-quality calibration set, the average PPL (30.52) is only slightly higher than the best result (30.29). This minimal degradation demonstrates the robustness of our method to calibration data quality.
>
> 3. **Stability Under Domain Mismatch**: Using domain-matched calibration data yields the best results (diagonal entries in Table 1). Even under domain mismatch, the increase in PPL is marginal. For example, calibrating with WikiText-2 on C4 results in a PPL of 19.28, only 0.03 higher than the domain-matched result (19.25). This confirms the stability of our method.
>
>
> >Q2: The paper briefly acknowledges that certain tasks, especially those requiring complex reasoning, might still be more affected by layer pruning, but does not provide a systematic analysis or mitigation strategies.
>
> A2: We thank the reviewer for highlighting this important point.
>
> * Currently, applying pruned models to complex reasoning tasks under training-free or lightweight training settings remains challenging. As shown in recent literature [1], pruning Llama-3.1-8B in a 2:4 format reduces its average sorting performance from 60.11 to 15.89. Similarly, our preliminary experiments using ShortGPT [2] to remove just three layers from DeepSeek-R1-Distill-Qwen-7B resulted in a significant drop in reasoning performance, with MATH-500 accuracy declining from 92.8 to 43.4.
>
> * We hypothesise that performance can be restored through retraining or constructing high-quality task-specific training data. Systematic analysis and mitigation strategies for complex reasoning tasks are currently unexplored and will be a key focus of our future work.
>
> ---
>
> [1] Srivastava, G., Cao, S., & Wang, X. (2025). Towards reasoning ability of small language models. arXiv preprint arXiv:2502.11569.
>
> [2] Men, X., Xu, M., Zhang, Q., Wang, B., Lin, H., Lu, Y., ... & Chen, W. (2024). Shortgpt: Layers in large language models are more redundant than you expect. arXiv preprint arXiv:2403.03853.
>
> >Q3: While the empirical results are strong, the paper does not provide theoretical analysis on why Hadamard transformation and scaling are sufficient for all types of activation mismatches.
> Q4: While the empirical benefits of Hadamard transformation and scaling are clear, can you provide more intuition or theoretical justification for why this combination is effective for aligning activation magnitudes after pruning? Additionally, are there identified scenarios where this approach fails or is suboptimal?
>
> A3: We thank the reviewer for highlighting the theoretical analysis on Hadamard transformation.
>
> * In the revised manuscript, we have added detailed explanation of theoretical analysis on Hadamard transformation in the Hadamard Transformation paragraph of Section 3.3. Briefly, multiplication with the Hadamard matrix encourages a more balanced distribution of activation across channels.
>
> * We provide mathematical proofs below.
> A symmetric Hessian matrix $H\in\mathbb{R}^{n\times n}$ is μ-incoherent if it has an eigendecomposition $H=Q\Lambda Q^{\top}$ such that for all $i$ and $j$, $|Q_{ij}|=|e_{i}^{\top}Qe_{j}|\leq\mu/\sqrt{n}$. One straightforward way to make a symmetric matrix incoherent is to conjugate it by a uniform random orthogonal matrix: this will result in each of its eigenvectors being a random unit vector, whose entries will concentrate around magnitude $n^{-1/2}$. [3][4]
>
> In this work, we introduce incoherence processing into activation with Hadamard transformation. A activation matrix $X\in\mathbb{R}^{n\times m}$is μ-incoherent if all $i$ and $j$,  $|X_{ij}|=|e_{i}^{T}We_{j}|  \leq\mu/||X||_F \sqrt{nm}$.
>
> Then, we obtain:
>
> $max(X) \leq\mu||X||_F /\sqrt{nm},$
>
> where $max$ is the element-wise max of the matrix, and $mn$ is the number of elements.
>
> Incoherence in activation $X$ can be viewed as a form of outlier reduction: a small bound on the magnitude of its entries across tokens means that we do not need to tailor a specific scaling parameter for each token and therefore eliminate the token-wise magnitude mismatch.
>
> ---
>
> [3] Ashkboos, S., Mohtashami, A., Croci, M. L., Li, B., Cameron, P., Jaggi, M., ... & Hensman, J. (2024). Quarot: Outlier-free 4-bit inference in rotated llms. Advances in Neural Information Processing Systems, 37, 100213-100240.
>
> [4] Chee, J., Cai, Y., Kuleshov, V., & De Sa, C. M. (2023). Quip: 2-bit quantization of large language models with guarantees. Advances in Neural Information Processing Systems, 36, 4396-4429.

---

> ### Author Response · Authors · 2025-08-07
> **Reminder**
>
> Dear Reviewer oSnA,
>
> I hope this message finds you well. As the discussion period is drawing to a close with less than two days remaining, I wanted to ensure we have addressed all your concerns satisfactorily, especially your insights about calibration data and theoretical analysis. If there are any additional points or feedback you'd like us to consider, please let us know. Your insights are invaluable to us, and we're eager to address any remaining issues to improve our work.
>
> Thank you for your time and effort in reviewing our paper.

---

### Official Review · Reviewer_xhnK · 2025-07-02

**Clarity:** 2
**Significance:** 3
**Originality:** 3
**Rating:** 4
**Confidence:** 4

**Summary:**

This paper proposes a novel framework to improve the performance of layer pruning in LLMs. This method is compatible with layer pruning methods that remove one or more consecutive layers. A patch, with a Hadamard transformation based initialization, is introduced at the location of the removed layers and then subsequently finetuned. Additionally, token-wise scaling is performed to bridge the gap between massive outliers at the different layers. The LinearPatch framework is compatible with layer pruning methods and outperforms the original versions of the layer pruning algorithms. The method is relatively efficient and lightweight, and a set of ablation studies demonstrate the contribution of each component to overall performance gains.

**Questions:**

N/A

**Ethical Concerns:**

["NO or VERY MINOR ethics concerns only"]

**Final Justification:**

The authors addressed my concerns and I would like to maintain my original score.

**Limitations:**

Yes

**Quality:**

3

**Strengths And Weaknesses:**

Overall I think that this is a good paper with a nice method. However, the writing needs to be improved and I would recommend copy editing of the entire paper.

Strengths:
- The method is compatible with existing layer pruning methods.
- The LinearPatch method outperforms baselines on a range of tasks.
- Ablation studies demonstrate the impact of each component of the pipeline.
- The main figure and description offer a clear presentation of the main method.

Weaknesses:
- The language throughout the entire paper needs to be improved. Several sentences are quite unclear and hinder understanding of key concepts. For example: "As a result, the pruned LLMs may suffer from efficient forward propagation as
35 before, which ultimately leads to the drop in performance." in lines 34-35 are awkward, use non standard terminology, and are unclear.

---

> ### Author Rebuttal · Authors · 2025-07-27
>
> >Q1: The language throughout the entire paper needs to be improved. Several sentences are quite unclear and hinder understanding of key concepts. For example: "As a result, the pruned LLMs may suffer from efficient forward propagation as 35 before, which ultimately leads to the drop in performance." in lines 34-35 are awkward, use non standard terminology, and are unclear.
>
> A1: We sincerely appreciate the reviewers' recognition of the plug-and-play LinearPatch method proposed in this paper, as well as the reviewer's careful observation and constructive feedback.
>
> * We thoroughly revise the entire manuscript, focusing on refining sentence structures, eliminating awkward phrasing, and ensuring consistent terminology usage.
>
> * Regarding the specific example cited in lines 34-35, the revised text now reads: "As a result, the pruned LLMs may suffer from $\textit{activation mismatch}$, which ultimately leads to the drop in performance." This revision clarifies the technical concept and uses standard terminology aligned with the field.

---

### Comment · Area_Chair_pfQN · 2025-08-01
**Reminder: Discussion Phase (July 31 – Aug 6)**

Hi everyone,

The Author-Reviewer Discussion phase is now open!

Please read the author responses, especially where you were mentioned, and post your initial reply as soon as possible. This helps ensure there's time for meaningful back-and-forth.

Thanks for your engagement!

-- AC

---

### Decision · Program_Chairs · 2025-09-17

**Decision:**

Accept (poster)

**Comment:**

This paper proposes LinearPatch, a lightweight and plug-and-play framework to mitigate activation mismatches in large language models subjected to layer pruning. The method combines Hadamard transformation, channel-wise scaling, and optional lightweight distillation to restore performance. It is pruning-metric agnostic and compatible with multiple pruning strategies, showing consistent improvements across various LLMs and benchmarks. Extensive ablation studies highlight the contributions of each component, and the approach achieves practical efficiency gains without requiring specialized hardware. The work identifies an important pathology in pruning and proposes a simple, deployable remedy with broad empirical validation.

During the review process, three reviewers provided positive scores (4, 4, and 5). Their main concerns have been addressed during the rebuttal. Reviewer 9aSJ raised several critical points, but many of them were also addressed by the authors (e.g., reproducibility, ablation, evaluation setup). Some of this reviewer’s additional demands (such as reproducing baselines from unspecified versions or providing links/emails outside the permitted review scope) are not reasonable within the review process.

Overall, the paper offers a technically sound, practically useful, and well-validated contribution to LLM pruning. Given the majority of positive reviews, the successful addressing of concerns, and the broad empirical validation of the approach, I recommend acceptance.